

# Hypoxia disaster in waters adjacent to the Changjiang estuary

Xiaofan Luo[1*], Hao Wei[2*], Renfu Fan[2], Zhe Liu[1], Liang Zhao[1]

[1] College of Marine Science and Engineering, Tianjin University of Science & Technology, Tianjin, 300457, China

[2] School of Marine Science and Technology, Tianjin University, Nankai District, Tianjin, 300072, China

*Correspondence to*: Hao Wei (weihao@ouc.edu.cn)

*[*]These authors contributed equally to this work and should be considered co-first authors.*

**Abstract.** Based on observational data from ten cruises carried out in 2012 and 2013, the distribution of dissolved oxygen (DO) and hypoxia (DO $<2.0$ mg L$^{-1}$) evolution in waters adjacent to the Changjiang estuary are studied. The linkage between

summer hypoxia and hydrodynamic conditions is explored. The results suggest that hypoxia frequently occurred from June to October to the south of the Changjiang estuary near the 30-50 m isobaths and was prone to happen under strong stratification without the presence of the Kuroshio Subsurface Water (KSW). Over the Changjiang Bank, hypoxia mainly occurred in July, August and September. Low-oxygen areas initially exist under strong stratification induced by the spreading of the Changjiang Diluted Water (CDW), and develop into hypoxia centers due to the lack of supplement of the relatively

DO-rich Yellow Sea Water and the KSW. The evolution of hypoxia in a year is influenced by conditions of the shelf circulation especially the paths of the KSW and the CDW. Thus, further study on the salinity evolution in the bottom layer of the water to the south of the Changjiang estuary and in the surface layer over the Changjiang Bank, that indicates the extensions of the KSW and the CDW, is needed for improving the hypoxia prediction.

## 1 Introduction

Dissolved oxygen (DO) is important for marine life. Low concentration of DO (e.g., $<3.0$ mg L$^{-1}$) causes hardship and even threats for most species living in the ocean. In coastal waters, increasing occurrence of hypoxia (DO $<2.0$ mg L$^{-1}$) is becoming a global environmental issue (Diaz and Rosenberg, 2008; Conley et al., 2009). One of the main goals of regional environment management is to reduce the area and volume of hypoxia, and this requires understanding and prediction of low-oxygen evolution (Feng et al., 2012). It is generally agreed that stratification and organic matter degradation are main

reasons for the formation of hypoxia. However, stratification is influenced by different physical processes; and for hypoxia in estuarine regions, there have been continuous debates on the roles played by the riverine nutrient loads and freshwater discharge (Bianchi et al., 2010).

Since the middle of last century, seasonal survey has started for hypoxia in the lower water column adjacent to the Changjiang estuary (Fig. 1), an area with a high primary production (Gu, 1980; Chen et al., 1988; Tian et al., 1993; Zhao et



al., 2001). Hypoxia was not identified as an important factor affecting the ecosystem of the East China Sea (ECS) until the study carried out by the Chinese GLOBEC (Global Ocean Ecosystem Dynamics) project in the summer of 1999 (Li et al., 2002). Subsequently, summer hypoxia in this region based on sparse ship-based observations has been reported (Shi et al., 2006a; Wei et al., 2007; Wang, 2009). The Changjiang estuary is currently regarded as one of the largest coastal hypoxia areas in the world (Chen et al., 2007). The spatial distribution and monthly evolution of low-oxygen adjacent to the

Changjiang estuary were revealed by analyzing the observations made monthly from May to October in 2006 on different station-grids by different research teams (Zhang et al., 2007; Zhou et al., 2010; Li et al., 2011; Wei et al., 2011; Wang et al., 2012). In 2006, low-oxygen center did not appear in May but existed in June in waters along the coast of Zhejiang Province (to the south of the Changjiang estuary). Over the Changjiang Bank, a hypoxia center appeared in the western part in July first, and then two hypoxia centers appeared to the north and south of the outer estuary in August, with the one in the north

being more severe. In September and October, the hypoxia center re-appeared to the south of the Changjiang estuary. Data from the monthly survey from September of 1958 to September of 1959 was recently re-analyzed (Liu et al., 2012). It was found that the low-oxygen center appeared to the south of the estuary in May and June, appeared near the mouth of the Changjiang River in July and August, and re-appeared to the south of the estuary in September. This evolution of the low-oxygen center showed correspondence with the extension of the Taiwan Warm Current (TWC). The lowest DO value was

0.34 mg L$^{-1}$ in August of 1959, a highly severe hypoxia but did not draw much attention for about 40 years.

Generally, the formation of hypoxia is related to high rate of primary production, long resident time of bottom water, persistent presence of stratification and specific bathymetry of continental shelf (Rabouille et al., 2008). For hypoxia in the waters adjacent to the Changjiang estuary, previous studies have identified the important roles played by the pycnocline between the freshwater from river discharge and the upwelled water from the TWC, and the abundant bottom detritus from

surface phytoplankton blooms (Li et al., 2002; Zhang et al., 2007). However, hypoxia did not always exist under the condition of pycnocline and mounting detritus, hence these processes cannot explain the evolution of the spatial distribution of hypoxia. Zhou et al. (2010) analyzed the observed bottom DO, water temperature and salinity off the Changjiang estuary in June, August, October of 2006, and made comparison with observational data in August of 1998 and 1999. They concluded that the temporal evolution of hypoxia is mainly caused by the spatial structure of water mass. Based on

observations in 2006, Wei et al. (2011) suggested that the northward spreading of hypoxia was caused by eutrophication in the Changjiang estuary, intensification of the TWC, and the northward shift of the upwelling region. Based on analysis of observed DO distributions in January and in each month from April to November in 2006, Wang et al. (2012) concluded that the location of hypoxia center was controlled by stratification and the northward expansion of the TWC. Zhang et al. (2012) suggested that stratification caused by the Changjiang Diluted Water (CDW) was the main reason for hypoxia formation

while the water advected from the Yellow Sea (YS) could weaken hypoxia; the competition of the two processes induced the spatial and temporal variations of DO concentration. On the basis of observations from June to August, and October in 2006



and from August to September in 2009, Zhu et al. (2015) emphasized the influence of pycnocline on the spatial variation of hypoxia. Through combining reanalysis of data from 2006 to 2007 with results of an ocean circulation model, Wei et al. (2015b) noted possible influence of seasonal circulation condition on the position of hypoxia off the Changjiang estuary.

Previous studies have reached a consensus that the movement of water masses and changes of stratification induced by ocean circulation variations are all responsible for the formation and development of hypoxia. However, there lacks agreement on which factor, among the TWC, the CDW, fronts and the Yellow Sea Water (YSW), plays the dominant role. It is possible that each previous study captured one aspect of hypoxia formation and variation in a specific area, but the conclusion was limited by the particular dataset analyzed. For instance, the hypoxia center in summer of 2006 was located at

a northern position relative to 1959, but did this northward shift occur frequently in recent years? Is it generally true that from beginning to the end of summer, the hypoxia center appeared successively at the southern, northern, then back to southern locations? What is the exact evolution of hypoxia from formation to decay? Besides the blocking of oxygen exchange by stratification, fronts and ocean eddies, how are DO distribution and hypoxia evolution influenced by the lateral transport of the TWC accompanied with the Kuroshio Subsurface Water (KSW)? What role does the YSW play?

From a synthesis of observations of hypoxia reported in literatures and made within the recent decade (Fig. 1 and Table 1), significant year-to-year variations in the location and evolution of hypoxia can be identified. This study attempts to further explore the influence of changes in hydrodynamic condition on the evolution of hypoxia through analyzing new observations carried out in June, August and October in 2012 and monthly from May to September in 2013. According to hydrodynamic condition and topography features, we divide the waters adjacent to the Changjiang estuary into the southern

and northern sub-regions (Fig. 1). The southern sub-region includes the offshore water of Zhejiang, the submarine river canyon and the middle shelf of the ECS. The northern region is the Changjiang Bank. The low-oxygen/hypoxia centers in the two years are mapped out from the observed bottom DO distribution. Distributions of stratification and hydrography are described. By linking the DO distribution to variations of hydrodynamic condition, we analyze the relative contributions of the TWC, the CDW and the YS tidal front in the two sub-regions, with a special attention paid on the influence of the lateral

advection of the KSW.

## 2 Field observations and data processing

Concentrations of DO and hydrographic data are obtained from 10 cruises carried out in June, August and October in 2012, and monthly from May to September in 2013 (Table 2). The data for May and July in 2013 is a merge of observations made from two ships in each month. RBR 620-CTD with Seapoint sensor of DO was used in all the 10 cruises to measure profiles

of water temperature, salinity, DO concentration. SBE-CTD was deployed in 8 cruises excluding the survey on Subei Shoal in 2013 and that within the Changjiang estuary in July of 2013. When both RBR 620-CTD and SBE-CTD took measurements, the profiles of water temperature and salinity from SBE-CTD were used because of the higher sampling



frequency and precision of this instrument.

Data recorded during descending profiling are used. For each profile, the inverse pressure calibration was applied and
spikes were removed as the basic data quality control. For each cruise that covered the whole observational grids, about 100
water samples from surface, middle and bottom layers were taken from stations along selected sections. Salinity of these
samples was measured by the SYA2-2 salinometer in laboratory to calibrate the salinity from the CTD measurement in each
cruise. The DO sensor on RBR620-CTD was calibrated prior to each cruise as the following. It was firstly put into pure
water aerated by oxygenate pump for 60 minutes to get the voltage for DO reaching 100 % saturation, and was then put into
the anoxia water (i.e., saturated sodium sulfite solution) to get the voltage for zero DO reading. The DO concentration was
determined according to the linear scaling between these two voltages. During some cruises, the chemistry group also used
the Winkler Titration method to measure the DO of water samples from different layers at selected stations. This enabled an
alternative calibration of the DO measured by the DO senor. Data was sampled at high frequencies, for instance, 24 Hz for
the SBE-CTD and 8 Hz for the RBR-620 CTD. The raw data was averaged over 0.2 m vertical intervals. The buoyant
frequency ($N^2$) was calculated from water temperature and salinity. The maximum $N^2$ in a profile was used to represent the
stratification intensity of the water column at a particular station.

Stations to the south of 33° N are chosen for analyzing the hypoxia off the Changjiang estuary (Fig. 1). Sampling grids
for each cruise and sections to be discussed in following sections are shown in Fig. 2. According to Su et al. (1996), salinity
can be used to identify water mass in the ECS, i.e., $S$ <30.0 representing the diluted water, 34.5> $S$ >34.0 for the TWC water,
and $S$ >34.5 for the KSW. The area with bottom temperature in the range of 13.0-17.0 ℃ stands for the summer tidal front of
the YS (Zhao, 1985). According to China's 1992 National Marine Investigation Standard by the State Bureau of Technical
Supervision, the phycnocline is defined to exist if the vertical gradient of density is larger than 0.1 kg m$^{-4}$ for the water depth
less than 200 m. This equals to the maximum $N^2$ of a vertical profile larger than $1 \times 10^{-3.0}$ s$^{-2}$.

## 3. Results

### 3.1. Bottom DO concentration in June, August and October, 2012

There was no hypoxia in June of 2012 (Fig. 3a). The bottom DO concentration was larger than 4.0 mg L$^{-1}$, with higher values
in the eastern part and lower values in the western part of survey area. A band of relatively low DO concentration of 4.0-5.0
mg L$^{-1}$ was found along the 30 m isobath to the north of 29° N.

In August, there were only a few stations located in the southern area (Fig. 3b). The DO concentration was 2.0-3.0 mg
L$^{-1}$ lower than that in June. Over the western side of the Changjiang Bank, i.e. near the Subei Bank, DO <3.0 mg L$^{-1}$ was
observed. A hypoxia center with DO = 1.0 mg L$^{-1}$ appeared in the northwestern survey field (station I1, at 33° N, 122.5° E).

In October, the DO concentration of bottom water was elevated (Fig. 3c). Two relative low-oxygen centers still existed





to the south of the Changjiang estuary though hypoxia disappeared. These two centers were located near the northern end of the submarine river canyon with DO = 4.8 mg L$^{-1}$ (station PN1), and the offshore waters of Zhejiang with DO <4.0 mg L$^{-1}$, about 2.0 mg L$^{-1}$ lower than that in the surrounding waters.

Overall, in summer of 2012 hypoxia off the Changjiang estuary was not severe. There was no hypoxia in June and October though the low-oxygen centers existed. The hypoxia center was located in the northern sub-region in August.

### 3.2. Bottom DO concentration from May to September, 2013

In May, the vertical and horizontal distributions of DO were fairly homogenous (Fig. 3d, vertical distribution was not shown). DO values were 8.0-9.0 mg L$^{-1}$ in the near shore region and 7.0-8.0 mg L$^{-1}$ in the offshore region. The minimum DO value, located at station kb, was 6.7 mg L$^{-1}$.

In June, DO was lower than 4.0 mg L$^{-1}$ in the southern sub-region. A hypoxia center appeared near the slope along the 30-50 m isobaths offshore of Zhejiang. DO values at most stations to the north of the Changjiang estuary were less than 5.0 mg L$^{-1}$ (Fig. 3e). Along the 122.5° E section, the DO concentration in the lower water column over the Changjiang Bank was substantially larger than that to the south of 31° N. In the southern area, the hypoxia water had a thickness of about 10 m in June (Fig. 4b). DO in June of 2013 was about 3.0 mg L$^{-1}$ lower than that in June of 2012 (Figs. 3a, e and Figs. 4a, b), and hypoxia occurred earlier in 2013. The occurrence of hypoxia in June was only reported in 2003 in the submarine river canyon near 30.45° N (Xu, 2005).

In July, bottom DO was larger than 7.0 mg L$^{-1}$ in the Changjiang estuary and Hangzhou Bay, except that DO <4.0 mg L$^{-1}$ was found at stations X5 and X6 near the northern corner of the Changjiang river mouth (Fig. 3f). Over the Changjiang Bank, at all stations with water depth larger than 30 m, bottom DO was less than 3.0 mg L$^{-1}$; severe hypoxia occurred within 32-32.4° N, 122.4-123.5° E over the western part of the bank. The huge area with DO <3.0 mg L$^{-1}$ extended to southwest of Jeju Island. DO values were 3.0-5.0 mg L$^{-1}$ over the middle shelf of the ECS. Overall, the DO concentrations in the southern and eastern parts of the survey area were higher than that in the middle part.

In August, the survey area was smaller than that in July. Over the Changjiang Bank, the area of hypoxia was still large, and extended northward and eastward compared with July (Fig. 3g). In the southern sub-region, a low-oxygen center with DO <3.0 mg L$^{-1}$ was located in the northern end of the submarine river canyon. DO distribution along section K (at 32° N) showed that the hypoxia water reached 30 m thickness from stations K2 to K6 over the Changjiang Bank, extending about 100 km in length (Fig. 4d). DO <2.0 mg L$^{-1}$ was observed at five stations and hypoxia area was separated into several patches extending northward to 33° N and eastward to 125° E. Over the whole bank, DO was less than 4.0 mg L$^{-1}$, much lower than that in August 2012 (Fig. 4c).

In September, the DO concentration over the western bank increased quickly, while DO <3.0 mg L$^{-1}$ was still found in the outer edge of the bank and hypoxia occurred near 33° N (Fig. 3h). Another area with DO <3.0 mg L$^{-1}$ was in the river canyon to the south of the Changjiang estuary.





In 2013, hypoxia first appeared in the southern sub-region in June and was sustained over the Changjiang Bank in July, August and September. Multiple low-oxygen centers appeared from June to September. In the southern sub-region, the hypoxia centers were stably located in the coastal water near the 30-50 m isobaths. In the northern sub-region, the positions of hypoxia centers changed around the bank. Overall, compared with historical reports, in the summer of 2013 hypoxia appeared earlier, occupied a larger area, with a larger thickness, and was maintained longer over the Changjiang Bank.

**4. Discussion**

**4.1 Climatological evolution of hydrodynamic conditions**

The evolution of hydrodynamic conditions in the Yellow and East China Seas is related to monsoon, Kuroshio, runoff of the Changjiang River, and bathymetry of the region (Su, 2001).

In each year, stratification starts to develop in May corresponding to surface heating. The tidal front in the YS appears
in May and gets intensified gradually in the following three months. This front can extend southeast-ward across the Changjiang Bank. In summer, the current along the western coast of the YS (within 20 m isobath) flows northward as a response to the wind (Liu and Hu, 2009). Both tidal front and northward coastal current are main obstacles for supplement of the YSW to the low-oxygen water over the Changjiang Bank.

The CDW extension is mainly influenced by the advancement and recession of the TWC, especially its inner branch
(Weng and Wang, 1985; Wei et al., 2015a). When the TWC intrudes further north over the continental shelf, the CDW tends to veer northeastward substantially. A strong halocline over the Changjiang Bank is created by the CDW and the water in lower layer. The combination of halocline and thermal stratification blocks the oxygenate aeration of the lower layer from the surface.

The TWC consists of Taiwan Strait water in the upper layer characterized with high temperature and moderately high
salinity, and the KSW in the lower layer with low temperature and high salinity (Weng and Wang, 1985). The pycnocline is formed in the region passing through by the TWC, and is also influenced by the coastal fresh water to the south of the Changjiang estuary. The KSW can be clearly identified by the characteristic oxygen concentration (4.5 mg L$^{-1}$). Hence the KSW provides the relatively low and high oxygen water before and after the formation of low-oxygen center (DO <3.0 mg L$^{-1}$), respectively. The TWC has two branches over the continental shelf. In May, the TWC is reinforced by the southerly
wind. The near shore branch of the TWC flows northward along the 50-60 m isobath offshore of Zhejiang into the submarine river canyon, and in July it can reach the northern corner of the Changjiang river mouth. This branch upwells to the sea surface in the coastal area to the south of the Changjiang estuary. The outer branch of the TWC can reach the Changjiang Bank along the middle shelf of the ECS. It flows along the edge of the bank in May and June, across the bank in July and August, along the edge again in September and October. Eventually, the TWC runs into the Tsushima Strait (Su, 2001).
In summary, the hydrodynamic condition, especially the TWC and the CDW, plays a leading role in the evolution,





duration and intensity of stratification in waters adjacent to the Changjiang estuary. According to laboratory experiments, Liu et al. (2012) proposed that the main reason for hypoxia formation was not oxygen consumption but the lack of replenishment after consumption. Hence hypoxia may occur in any area if stratification is sustained over a considerable duration without DO replenishment.

**4.2. Bottom DO, KSW intrusion and stratification in the southern sub-region**

In the southern sub-region, the appearance of low-oxygen centers was in consistent with the distribution of strong stratification (Fig. 5), and the evolution of DO concentration can be influenced by the lateral transport of the KSW with DO = 4.5 mg L$^{-1}$.

A low-oxygen center often first occurs in the confluent area of coastal current and the TWC, which is usually located 195 between 30 m and 50 m isobaths where stratification is stronger and more persistent than that in the middle shelf of the ECS. In June and August of 2012, because of the intensive KSW intrusion toward the northern corner of the mouth of the Changjiang (Figs. 6a, b), hypoxia did not happen despite of the presence of strong stratification ($N^2 > 10^{-2.5}$ s$^{-2}$) (Figs. 5a, b). In October of 2012, owing to the TWC recession and enhanced monsoon, stratification was overall weak and DO increased (Figs. 3c, 5c, 6c and Table 3).

From May to September of 2013, the southern coastal area was strongly stratified (Figs. 5d-h and Table 3). In May, the persistence of stratification was insufficient for oxygen consumption, resulting the presence of relatively high DO concentration. In June, the southern branch of the CDW expanded more widely relative to June 2012 (Figs. 7a, e), and the TWC reached the latitude of 30.5° N (Fig. 6e). This led to a large $N^2$ values ($N^2 > 10^{-2.0}$ s$^{-2}$) in the coastal water to the south of the Changjiang estuary (Fig. 5e). With the presence of stratification, DO was rapidly consumed and a large low-oxygen area 205 formed. In the meanwhile, the KSW was located to the south of 28° N (Figs. 6e, 8e) and could not provide the relatively DO-rich water via lateral transportation. As a consequence, the low-oxygen area developed into hypoxia (Fig. 3e). In July, with the KSW intruding northward and occupying the majority of bottom water in the southern sub-region (Fig. 6f), hypoxia faded away and did not re-appear afterward. Thus, the timely replenishment of the KSW could prevent the evolution of low-oxygen center into hypoxia.

The scatter plot of the bottom salinity versus DO concentration in the southern sub-region (Fig. 9a) shows that hypoxia did not occur when $S > 34.5$; and when hypoxia occurred, $S$ must be less than 34.5. This is consistent with our above analysis. Evidences supporting this conclusion can be found in previous studies. Hypoxia occurred in the coastal area to south of the Changjiang estuary in August of 1959, 1976-85, 1981, 1999, 2002 and 2006; June of 2003; and September and October of 2006 (Table 1). Among these years, in 2006 hypoxia was maintained for three months while the KSW intruded at a southern 215 location (Zou et al., 2008; Zhou et al., 2010; Wang et al., 2012), and in August 1959 hypoxia occurred in the northern end of the canyon where bottom salinity was 33.0 (Liu et al., 2012). No input of DO-rich water from upstream led to the generation of hypoxia in this sub-region. For the rest hypoxia cases, there was insufficient hydrological data to identify the locations of



the KSW intrusion.

### 4.3. Bottom DO, CDW spreading and stratification in the northern sub-region

In the northern sub-region, the low-oxygen centers were located where stratification was strong, and the pattern of hypoxia was influenced by the extension of the CDW.

In June 2012, both strong stratification and relative low-oxygen occurred in the western part of the Changjiang Bank (Figs. 3a, 5a); and till August, strong stratification was maintained and DO decreased to less than 3.0 mg L$^{-1}$ (Figs. 3b, 5b). The CDW veered northeastward substantially (Fig. 7b), and below it a salinity front was produced as the TWC encountered

the coastal fresh water (Fig. 6b). To the east of salinity front, there existed a tidal front (Fig. 8b). The DO-rich YSW was transported to the eastern bank along this tidal front. Due to the lack of DO replenishment, the low-oxygen area to the north of salinity front in the western bank developed into hypoxia (Fig. 3b). In October, the CDW switched back to flowing southward along the coast of Zhejiang. As stratification disappeared ($N^2 < 10^{-3.5}$ s$^{-2}$), bottom DO increased to 7.0 mg L$^{-1}$ (Figs. 3c, 5c).

In summer of 2013, the low-oxygen center and hypoxia in the bank all showed correspondence with strong stratification ($N^2 > 10^{-2.0}$ s$^{-2}$) (Fig. 3 and Fig. 5). In July, the CDW expanded eastward, covering a zone spanning from the Changjiang estuary to Jeju Island. This facilitated the formation of a large low-oxygen area with DO <3.0 mg L$^{-1}$ over the whole bank under strong stratification (Figs. 3f, 5f, 7f). In the meanwhile, the KSW and the YSW along the tidal front relieved the hypoxia in the southern and eastern bank, respectively (Figs. 6f, 8f). Hence, hypoxia events only happened in the western

bank without DO supplement (Fig. 3f). In August, the KSW did not approach the mouth of the Changjiang River, but retreated southward (Fig. 6g). Meanwhile, the northward migration of the tidal front (Fig. 8g) limited the DO replenishment by the YSW. As a consequence, a sever hypoxia evolved in the area with low-oxygen and sustained stratification (Fig. 3g, 5g and Table 3). In September, over the Changjiang Bank the CDW shrunk, stratification was broken almost everywhere (Figs. 5h, 7h), and DO increased accordingly (Fig. 3h and Table 3). At the outer edge of the bank, however, stratification ($N^2 > 10^{-3.0}$

s$^{-2}$) was still present because of the vertical temperature differences, low DO (<3.0 mg L$^{-1}$) was maintained along the outer edge (stations I4, I5 and K6), and hypoxia occurred at the northeastern location (station I5).

The bottom DO concentration was in negative correlation with the strength of stratification over the Changjiang Bank (Fig. 9b). A regression relationship was obtained as DO = -1.67 × Log$_{10}N^2$ + 0.43,with r = -0.67 (significant at the 0.05 confidence level). Values of DO were always larger than 4.0 mg L$^{-1}$ when stratification was very weak ($N^2 < 10^{-3.0}$ s$^{-2}$).

Hypoxia only happened under strong stratification, though high DO concentration could also exist under the same situation. Thus, stratification is a necessary condition for hypoxia formation over the bank.

In summer, the CDW will turn onto the bank but the direction of its extension may vary. For instance, the CDW spread northward in June 2012, and eastward in July 2013. The largest values of $N^2$ were reached when the CDW lay over salty waters. Overall, hypoxia usually formed in the northwestern Changjiang Bank when the CDW extended northwestward, and



in the eastern part when the CDW spread eastward substantially.

**4.4. Comparison of hypoxia development between the water adjacent to the Changjiang estuary and the Gulf of Mexico**

Worldwide, the frequent occurrence of hypoxia is related to eutrophication (Conley et al., 2009). For Gulf of Mexico, a statistical model for hypoxia prediction has been established based on nutrient loads from Mississippi River (Turner et al.,
2006). For Chesapeake Bay, the prediction of hypoxia volume is related to river runoff (Scully, 2010). However, for waters adjacent to the Changjiang estuary, hypoxia can be influenced by many complicated factors. Here hypoxia is not directly related to runoff from river. In summer of 2006 the runoff was extremely low (Fig. 10) but severe hypoxia occurred (Zou et al., 2008; Zhou et al., 2010; Wang et al., 2012). In summer of 2012 the runoff was among the highest in the recent decade but only a small area of hypoxia existed in August (Figs. 3a-c and Fig. 10). Compared with 2012, in 2013 the runoff was far
more less but hypoxia occupied extremely large areas and sustained much longer (Figs. 3d-h and Fig. 10). Clearly, the area or volume of hypoxia adjacent to the Changjiang estuary cannot be predicted based on river runoff.

Over the Texas-Louisiana shelf in the northern Gulf of Mexico, the formation and destruction of hypoxia is primarily a local vertical process, and the biological processes are responsible for producing hypoxia change from east to west (Hetland and DiMarco, 2008; Bianchi et al., 2010). Hypoxia is predominantly caused by water column respiration in the eastern
region with steep shelf break, and by benthic respiration over the wide shelf to the west. Similarly, the hypoxia area adjacent to the Changjiang estuary can be divided into an area on the wide bank and an area with steep slope according to bathymetry. Over the Changjiang Bank, low-oxygen is related to the presence of the CDW, while the CDW's position is influenced by the TWC (Wei et al., 2015a). Strong tidal mixing and monsoon may both induce the detachment of freshwater from the CDW (Xuan et al., 2012). This results in stronger changes of the river plume on the Changjiang Bank, compared with that on the
west part of the Texas-Louisiana shelf where tides and wind are relatively weak. Although the low-oxygen centers to the south of the Changjiang estuary are constrained by bathymetry, the bottom DO concentration is significantly influenced by the shelf circulation including the TWC and shoreward intrusion of the Kuroshio. The lateral transport and vertical processes both control the hypoxia adjacent to the Changjiang estuary. The prediction of low-oxygen needs information about salinity evolution at the bottom to the south of the Changjiang estuary and at the surface over the Changjiang Bank, that defines the
extensions of the KSW and the CDW, respectively.

**5. Conclusions**

As one of the major causes of ecological disasters in the coastal seas, especially in large river estuaries, hypoxia is becoming a global environmental issue and has attracted wide attention in recent decades. Base on new observational data from 10 cruises carried out in 2012 and 2013, the distribution of dissolved oxygen and evolution of hypoxia in waters adjacent to the
Changjiang estuary are studied. The linkage of summer hypoxia with hydrodynamic conditions, including variations of water



mass and stratification associated with circulation, is explored. The study area can be divided into the southern (south of Changjiang estuary) and northern (the Changjiang Bank) sub-regions. The mechanisms dominating the evolution of stratification and DO distribution are different in these two regions.

a) In the southern sub-region, a confluent area of coastal current and TWC near the 30-50 m isobaths is where a low-oxygen center usually exists, and hypoxia may occur from June to October accompanied by strong stratification. Associated with bathymetric feature, fronts exist throughout of the year in the northern end and block the DO exchange with water to the north. Thus lateral transportation is mainly from the south. The KSW ($S$ >34.5) carried by the near shore branch of the TWC can substantially affect the bottom DO concentration in this region. Hypoxia is difficult to form if the KSW appears early in the middle shelf of the ECS and intrudes further northward in summer. In this region, hypoxia does not occur at locations with bottom $S$ >34.5; and when hypoxia does occur, there must be bottom $S < 34.5$.

b) In the northern sub-region, i.e. over the Changjiang Bank, there is a high possibility that hypoxia occurs from July to September. The distribution of bottom DO corresponds to the intensity of stratification, the two being in negative correlation with a regression relationship of DO = $-1.67 \times \mathrm{Log}_{10}N^2 + 0.43$. DO is larger than 4.0 mg L$^{-1}$ when stratification is weak ($N^2$ <10$^{-3.0}$ s$^{-2}$). Over the bank, low-oxygen area is mostly related to the CDW spreading. The combination of the Subei coastal current (northward in summer), tidal front of the YS and the bathymetry of the bank restrains DO replenishment from surrounding waters. Under the condition of sustained stratification, a low-oxygen center may involve into hypoxia. Generally, the coastal current transports the YSW to the Changjiang Bank after the transition of monsoon in autumn, and in the meanwhile stratification has already disappeared. Thus, water from the YS may play a minor role in the relief of hypoxia in summer.

In summary, based on new observations in 2012 and 2013, especially in 2013 with monthly data from May to September, it is evident that hypoxia in waters adjacent to the Changjiang estuary is substantially influenced by shelf circulation, including the CDW plume, the TWC and the KSW intrusion. The hypoxia evolution in 2013 was distinctly different from that in 2006 and 1958-1959. Due to the lack of monthly observation covering the same region in different years, credible model simulations of inter-annual variations of hydrodynamic conditions are needed for further investigating the linkage of hypoxia with the KSW intrusion and the CDW spreading. It is hopeful that the prediction of hypoxia can be made according to the evolution of surface and bottom salinity.

**Acknowledgements**

This study was supported by the National Basic Research Program of China (Grant No.2011CB403606 & 2010CB428904) and the National Natural Science Foundation of China (NSFC, Grant No.41376112). Liang Zhao was funded by NSFC under Grant No. 41276016 and the Strategic Priority Research Program of the Chinese Academy of Sciences (XDA11020305). Dr. Youyu Lu is appreciated for helpful discussions.



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





## Tables and Figures:

**Table 1.** Records of locations and values of minimum dissolved oxygen off the Changjiang estuary.

| Time | Location | | Min. DO value (mg L⁻¹) | References |
|---|---|---|---|---|
| | Longitude | Latitude | | |
| Aug., 1959 | 122°45′ | 31°15′ | 0.34 | Liu et al., 2012 |
| Aug., 1976–1985 | 123°00′ | 31°00′ | 0.80 | Zhang, 1990 |
| Aug., 1981 | 123°00′ | 30°50′ | 2.0 | Limeburner et al., 1983 |
| Aug., 1988 | 123°00′ | 30°50′ | 1.96 | Tian et al., 1993 |
| Aug., 1998 | 124°00′ | 32°10′ | 1.44 | Wang and Wang, 2007 |
| Aug., 1999 | 122°59′ | 30°51′ | 1.0 | Li et al., 2002 |
| Aug., 2002 | 122°29′ | 32°00′ | 1.73 | Shi et al., 2006a, 2006b |
| | 123°00′ | 31°00′ | 1.99 | |
| Jun., 2003 | 122°50′ | 30°50′ | ~1.0 | Xu, 2005 |
| Aug., 2003 | 123°30′ | 31°30′ | 2.0 | Chen et al, 2007 |
| Sep., 2003 | 122°56′ | 30°49′ | 0.8 | Wei et al, 2007 |
| | 122°45′ | 31°55′ | <1.5 | |
| Aug., 2005 | 122°48′ | 32°18′ | 1.55 | Zhu et al, 2007 |
| Jul., 2006 | 122°23′ | 32°42′ | 1.36 | Wei et al, 2010 |
| Aug., 2006 | 122°55′ | 32°55′ | 1.0 | Zhou et al, 2010 |
| | 122°30′ | 29°12′ | <2 | |
| Sep., 2006 | 123°06′ | 30°09′ | 1.96 | Zou et al., 2008 |
| Oct., 2006 | 123°21′ | 30°05′ | 1.8 | Zhou et al, 2010 |
| Aug., 2012 | 122°41 | 33°00′ | 1.0 | this study |
| Jun., 2013 | 122°39′ | 29°50′ | 1.5 | this study |
| | 122°52 | 29°48′ | 1.6 | |
| Jul., 2013 | 122°19′ | 32°24′ | 1.7 | this study |
| | 122°19′ | 32°03′ | 1.6 | |
| | 122°36′ | 31°54′ | 1.8 | |
| | 123°25′ | 32°10′ | 1.9 | |
| Aug., 2013 | 123°44′ | 33°00′ | 1.8 | this study |
| | 123°01′ | 32°00′ | 1.9 | |
| | 123°32′ | 32°00′ | 1.8 | |
| | 124°28′ | 32°00′ | 1.3 | |
| | 124°46′ | 32°00′ | 1.2 | |
| Sep., 2013 | 123°58′ | 33°02′ | 1.1 | this study |





**Table 2.** Information of cruises analyzed in this study.

| Cruise | Sampling Date | Number of Stations and Research Area | Instrumentation | Research Vessel |
|---|---|---|---|---|
| 201206 | 08-29 Jun. | 54 stations in the YECS[a] | SBE 17*plus* RBR620 | *Beidou* |
| 201208 | 11-20 Aug. | 42 stations in the YECS | SBE 17*plus* RBR620 | *Beidou* |
| 201210 | 09-26 Oct. | 42 stations in the YECS | SBE 19*plus* RBR620 | *Beidou* |
| 201305-I | 11-15 May | 9 stations in Subei Bank | RBR620 | *Suruyuyun288* |
| 201305-II | 11-15 May | 15 stations off Zhejiang Coast | SBE 911*plus* RBR620 | *Zhepuyu 23012* |
| 201306 | 15-19 Jun. | 48 stations in the YECS | SBE 911*plus* RBR620 | *Beidou* |
| 201307-I | 16-22 Jul. | 20 stations in the Changjiang estuary | RBR620 | *Runjiang1* |
| 201307-II | 12 Jul. - 01 Aug. | 74 stations in the YECS | SBE 911*plus* RBR620 | *Dongfanghong2* |
| 201308 | 13 Aug. - 02 Sep. | 63 stations in the YECS | SBE 17*plus* RBR620 | *Beidou* |

[a] YECS: Yellow and East China Seas




**Table 3.** Regional mean temperature at the bottom ($T_{bottom}$: in ℃), salinity in the surface ($S_{surface}$: in psu), salinity at the bottom ($S_{bottom}$: in psu), maximum squared buoyancy frequency at logarithmic scale ($N^2_{max}$: in s$^{-2}$), dissolved oxygen concentration of bottom water (DO: in mg L$^{-1}$) and the number (N) of samples in areas with frequent occurrence of hypoxia.

|  | Frequent hypoxia area | $T_{bottom}$ | $S_{surface}$ | $S_{bottom}$ | $N^2_{max}$ | DO | N |
|---|---|---|---|---|---|---|---|
| 201206 | I [a] | 18.9±1.6 | 31.5±3.2 | 34.9±0.7 | -2.1±0.2 | 5.2±0.8 | 14 |
| 201208 | I | 18.7±0.2 | 29.8±4.2 | 35.7±0.0 | -1.9±0.2 | 3.5±0.5 | 2 |
| 201210 | I | 23.0±0.9 | 30.4±3.9 | 31.9±1.7 | -2.5±0.6 | 5.3±1.7 | 5 |
| 201305 | I | 17.4±0.8 | 25.9±3.9 | 28.6±5.2 | -2.8±0.4 | 8.1±0.3 | 10 |
| 201306 | I | 19.1±1.1 | 28.7±4.2 | 33.9±0.8 | -2.0±0.3 | 2.7±0.7 | 11 |
| 201307 | I | 20.4±2.1 | 31.2±3.4 | 33.0±2.5 | -2.0±0.6 | 4.8±1.7 | 19 |
| 201308 | I | 10.0±1.4 | 31.3±3.8 | 34.2±0.4 | -2.3±0.3 | 3.6±0.8 | 7 |
| 201309 | I | 21.1±2.7 | 32.4±1.6 | 34.0±0.9 | -2.1±0.1 | 4.4±1.2 | 6 |
| 201206 | II [b] | 16.9±2.4 | 29.5±2.9 | 33.0±0.8 | -2.2±0.9 | 6.4±1.5 | 13 |
| 201208 | II | 20.5±3.9 | 28.1±5.2 | 33.6±1.9 | -1.9±0.4 | 3.7±1.4 | 11 |
| 201210 | II | 21.7±0.8 | 30.9±0.8 | 31.0±0.8 | -3.8±0.4 | 7.2±0.4 | 14 |
| 201305 | II | 15.2±1.2 | 27.2±4.2 | 31.7±1.1 | -2.7±0.9 | 8.3±0.7 | 14 |
| 201306 | II | 17.2±3.0 | 30.2±1.5 | 31.7±0.7 | -2.5±0.5 | 4.3±0.7 | 12 |
| 201307 | II | 20.5±2.5 | 29.6±1.5 | 31.9±1.1 | -1.7±0.2 | 2.8±0.8 | 11 |
|  | ($S_{surface}$ >26.0) |  |  |  |  |  |  |
|  | II | 27.5±4.2 | 7.7±8.5 | 10.0±13.6 | -2.5±1.0 | 5.1±2.7 | 5 |
|  | ($S_{surface}$ <26.0) |  |  |  |  |  |  |
| 201308 | II | 22.1±3.0 | 31.2±1.9 | 32.7±0.9 | -2.3±0.7 | 2.2±0.7 | 15 |
| 201309 | II | 23.6±2.4 | 31.6±2.3 | 32.9±1.0 | -2.9±0.8 | 5.0±2.0 | 15 |

[a] offshore waters of Zhejiang Province and submarine river canyon

[b] over the Changjiang Bank with water depth less than 50 m





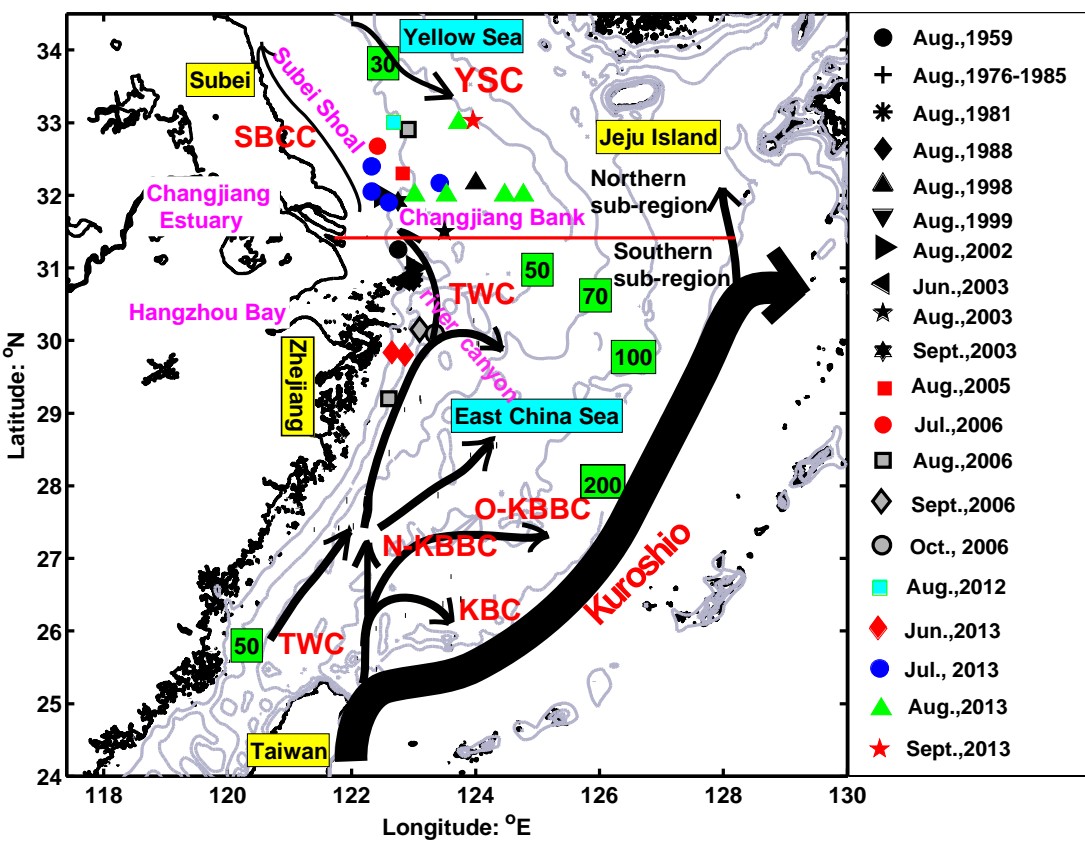

**Figure 1.** Topography and schematic map of summer circulation in the Yellow Sea and East China Sea. Black arrows denote circulation including the Yellow Sea Current (YSC), Subei Coastal Current (SBCC), Taiwan Warm Current (TWC), and Kuroshio intrusion (KBC: surface Kuroshio Branch Current; O-KBBC: offshore Kuroshio Bottom Branch Current; N-KBBC: Nearshore Kuroshio Bottom Branch Current, i.e. Kuroshio Subsurface Water). Contours in gray denote isobaths of 30, 50 70, 100, and 200 m. Various symbols represent sampling stations of different cruises labelled in the right pane that observed the occurrence of hypoxia. Red line marks the boundary between southern and northern sub-regions.



**Figure 2.** Maps of sampling stations off the Changjiang estuary (a-c: June, August and October in 2012; d-h: May to September in 2013). The layout is designed to ease comparison between 2012 and 2013 for June and August. Black, green and blue bold lines denote section at 122.5°E and section K in Fig. 4, and section PN in Fig. 5, respectively.





**Figure 3.** Dissolved oxygen concentration (mg L$^{-1}$) of bottom water in the same layout as Fig. 2.





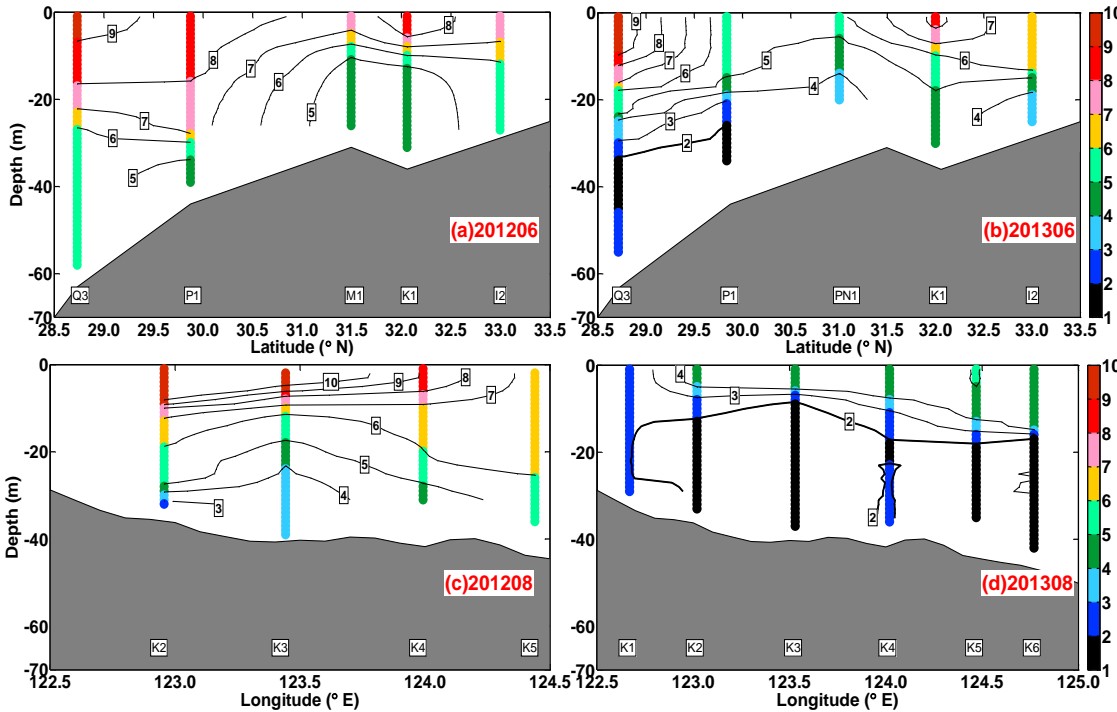

**Figure 4.** Dissolved oxygen concentration (mg L$^{-1}$) along section at 122.5°E (a: June 2012; b: June 2013) and section K (c: August 2012; d: August 2013).





**Figure 5.** Same as Fig. 3 except for maximum squared buoyancy frequency at logarithmic scale ($N^2$: s$^{-2}$).






**Figure 6.** Same as Fig. 3 except for bottom salinity (in psu). The contour of $S = 30.0$ in blue represents the extension of the Changjiang Diluted Water; $S = 34.0$ in magenta represents the Taiwan Warm Current; and $S = 34.5$ in red indicates the intrusion of Kuroshio Subsurface Water.





**Figure 7.** Same as Fig. 6 except for surface salinity (in psu).




**Figure 8.** Same as Fig. 3 except for bottom temperature (in °C). The 18°C contour characterizes the Kuroshio Subsurface Water and is emphasized in bold. Shading area denotes frontal zone.






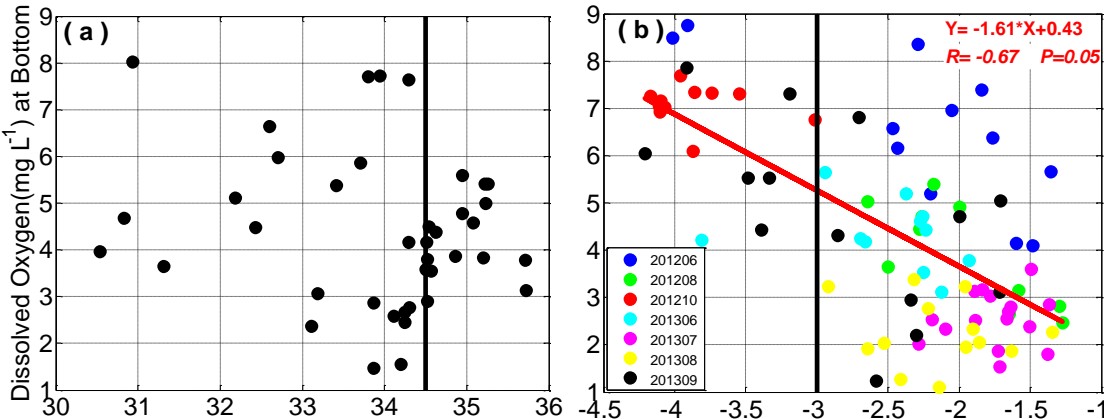

**Figure 9.** Dissolved oxygen concentration (in mg L$^{-1}$) of bottom water versus (a) bottom salinity (in psu) in the southern sub-region, and (b) maximum squared buoyancy frequency at logarithmic scale ($N^2$: s$^{-2}$) in the northern sub-region.




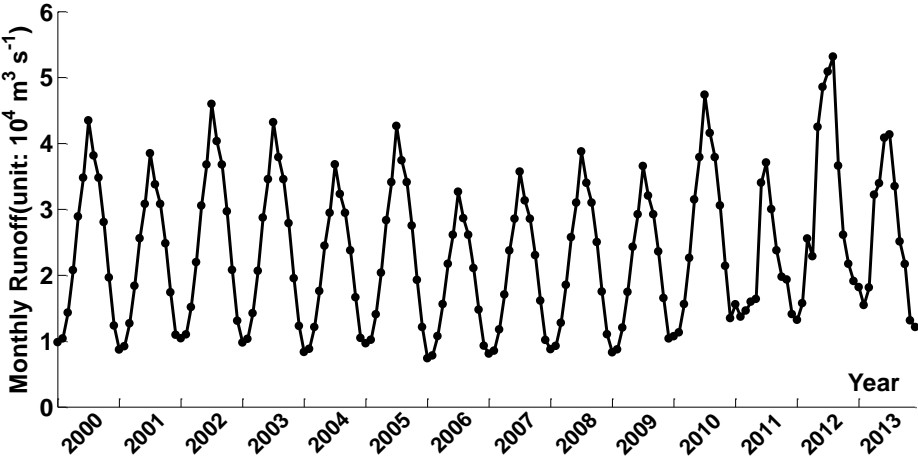

**Figure 10.** Monthly discharge of the Changjiang River in 2000–2013, averaged from daily monitoring data collected at the Datong Hydrologic Station (http://yu-zhu.vicp.net/).
