# Peer review of "Hypoxia disaster in waters adjacent to the Changjiang estuary"

_Natural Hazards and Earth System Sciences, 2016_

## Referee Comment (RC1) · Anonymous Referee #1 · 24 Jun 2016

Review of the manuscript: Hypoxia disaster in waters adjacent tio the Changjiang estuary X. Luo, H. Wei, Z. Liu, L. Zhao Submitted for publication on Naturla Hazards and Earth System Science. Ref: 2016-59

The manuscript is based on a series of hydrological surveys in the region adjacent and offshore the Changjiang estuary, an area that suffers from frequent hypoxia/anoxia episodes. The background information, provided by the Authors in the introductory part of the manuscript, inform the reader that hypoxia development is due to the classical triggering factors: enhanced vertical stratification and organic matter accumulation and remineralsation in the lower water column, but they state that the timing of the hypoxia onset, as well as the location of the main hypoxia center is deterrmined by the interplay between the water masses of the region. The main effort of the paper is about an effort to define the spatial and temporal hydrological characteristics underlying the

development of the hypoxia events as well as their location.

I have to say that despite the rich and extensive dataset assembled, the effort is not successful because of a rather confusing analysis and description of the data collected. Therefore I do not recommend the publication of the manuscript in its present form. Below I list a series of remarks that hopefully the authors might consider in addressing the very serious major revision that the manuscript needs in order to be considered for publication in future.

1) The manuscript reads very much as a technical report rather than as a scientific paper. There is a long description of the paper figures that unfortunately does not help very much the reader to understand the following considerations.

2) The hydrology of the region (Water mass distribution and pathways of the main currents) is described by means of a qualitative cartoon only. However in the following the distribution of the observed hydrological properties (T S and DO) is related to specific water masses. Unfortunately the reader not knowledgeable with the oceanographic characteristics of the region, cannot fully understand and assess the dependence of the hypoxia onset and location on the basis of the changing hydrologiy. It is therefore strongly recommended that the authors define more strictly the hydrological properties of the water masses involved in the hypoxia dynamics of the region (the large quantity of data they collected should enable them to provide (for instance) T-S diagram whose analysis can help to define in a quantitative way the interplay among water masses. Also the use of T-DO and/or S-DO diagrams could greatly help the analysis and the considerations about hypoxia timing and location

3) The figure acompanying the manuscript are very poor and confusing. Again the large quantity of data they collected should deserve a better analysis, based (for instance) on an objective analysis procedure, who would allow the author to define better the location of the water masses. From a formal point of view the combined use of isolines and "colored" dots is adding confusion.

4) As it is now it is very much difficult to relate the main conclusion of the paper with the data described and analysed earlier. This is due , as stated above, to the poor treatment of the data and the generic analysis procedure.

---

## Referee Comment (RC2) · Anonymous Referee #2 · 25 Jul 2016

Based on a new and extensive dataset (cruises) the authors intent to relate the dynamics of hypoxic area (formation, spatial distribution, timing) in the Changjiang Estuary to local hydrodynamics features.

The strong hypoxic event documented highlights the scientific relevance of the issue.

The most important conclusion is that the interannual variability of hypoxic events in this region is primarily driven by physical processes rather than riverine run off or nutrient input (see comparison with Gulf of Mexico in Sect 4.4) and the details provided for this control. The authors details those physical process for two distinct regions, i.e 1. the southern estuary, where hypoxia is avoided in cases of Kuroshio Subsurface water northward extension and 2. the Changjiang Bank (or northern estuary) where hypoxia occurrence is related to persistent (haline) stratification triggered by the westward spreading of Changjiang diluted waters. Those conclusions are well supported by the summarizing Fig. 9. and form a understanding basis for a more extensive reading of local datasets.

The analysis is well documented and instructed, in the sense that observations are described and discussed in the lights of know dynamics of the region rather than passed through automatic statistical procedures. In my view this is positive given the complexity of local circulation features.

The abstract gives a clear summary of the results and conclusions.

I advise the publication of this manuscript, after considering of the few minor revisions listed below.

Major comments.

Sect 4.4 I think the discussion should include comments on the different timescales considered. Biogeochemical and physical drivers does not play on the interannual variability of hypoxia occurrences at the same time scales. Another issue is that the gulf of Mexico and the East China Sea differ by their openess, which might changes the relative importance of physical and biogeochemical drivers.

Minor Comments.

L62. "the influence of pycnocline on the spatial variation of hypoxia" -> the relationship between the pycnocline ?location? and spatial variation of hypoxia ?

L70 extra space in "t hat"

L73 Can you give reference for the blocking of oxygen exchanges by ocean eddies ? if not, remove.

L103 senor -> sensor

L111 references needed for the "investigation standards"

L112 phycnocline -> pycnocline

L148. Either "reached 30m at station K3", or " reached 20-30 m from stations K2 to K6"

L166.and Fig 8 can you give more references or the method used to locate the frontal zone.

L172 oxygenate aeretion -> oxygenation or ventilation

L176. Reference for characteristic oxygen concentrations of the KSW.

L184 Tsushima strait is not on the map Fig1.

L191. "was consistent"

L201 rephrase "the persistence of stratification was insufficient for oxygen consumption"

L204 "DO was rapidly consumed" -> "was rapidly depleted".

L216 rephrase: "The absence of DO-rich water input from upstream led to .. "

L217 "rest" -> "other"

L243 remove the uppercase from Log and put $N^2$ inside ()

L259 "was far more less" -> "was lower by far"

L273 needs -> requires

L286 remove "of"

L296 involve -> evolve

L305 last sentence has to be rephrased to something like "Our conclusions could support the prediction of hypoxia based on the evolution of bottom and surface salinity"

References : Chen et al is 1988 on L.29 and 1998 in the references; Zhao et al 1985,

not reffered to in the manuscript

---

## Author Comment (AC1) · 15 Aug 2016

Response to Review

We thank both reviews for very constrictive comments and suggestions. These are very helpful in guiding our revisions of the manuscript. We make the efforts to thoroughly revise in order to improve the presentation of our data analysis and the interpretation of results. A point-to-point response to comments is provided below. Reviewers' comments are in blue font and our response is in black font. Please also note that we have changed the title of this manuscript to "On influencing factors of hypoxia in waters adjacent to the Changjiang estuary" and added an additional author (Youyu Lu).

Reviewer 1: The manuscript is based on a series of hydrological surveys in the region adjacent and offshore the Changjiang estuary, an area that suffers from frequent hypoxia/anoxia episodes. The background information, provided by the Authors in the introductory part of the manuscript, inform the reader that hypoxia development is due to the classical triggering factors: enhanced vertical stratification and organic matter accumulation and remineralsation in the lower water column, but they state that the timing of the hypoxia onset, as well as the location of the main hypoxia center is determined by the interplay between the water masses of the region. The main effort of the paper is about an effort to define the spatial and temporal hydrological characteristics underlying development of the hypoxia events as well as their location. I have to say that despite the rich and extensive dataset assembled, the effort is not successful because of a rather confusing analysis and description of the data collected. Therefore, I do not recommend the publication of the manuscript in its present form. Below I list a series of remarks that hopefully the authors might consider in addressing the very serious major revision that the manuscript needs in order to be considered for publication in future.

R: We thank this reviewer for the very critical review that points out the weakness in our analysis. The detailed comments are fully considered in our revision. (See our response below)

1) The manuscript reads very much as a technical report rather than as a scientific paper. There is a long description of the paper figures that unfortunately does not help very much the reader to understand the following considerations.

R: Following the reviewer's comment, we have made significant effort to simply description of observational data (figures) and enhance the statistical analysis and interpretation.

2) The hydrology of the region (Water mass distribution and pathways of the main currents) is described by means of a qualitative cartoon only. However, in the following the distribution of the observed hydrological properties (T S and DO) is related to specific water masses. Unfortunately the reader not knowledgeable with the oceanographic

characteristics of the region, cannot fully understand and assess the dependence of the hypoxia onset and location on the basis of the changing hydrology. It is therefore strongly recommended that the authors define more strictly the hydrological properties of the water masses involved in the hypoxia dynamics of the region (the large quantity of data they collected should enable them to provide (for instance) T-S diagram whose analysis can help to define in a quantitative way the interplay among water masses. Also the use of T-DO and/or S-DO diagrams could greatly help the analysis and the considerations about hypoxia timing and location.

R: Following this very insightful and constructive suggestion from the reviewer, we have made substantial effort to improve the analysis of relationship between water mass and DO distributions. First, Figure 1 has been revised to denote the major branches of summer circulation that contribute to the shaping of hydrologic conditions in the region. We divide the focused study area into regions I and II, to simplify the description in the main text and ease the appreciation by a reader who is unfamiliar with the regional oceanography. We further add a new Figure 9 to explore the variations of DO with respect to T, S and N2. These new diagrams show clear grouping of observational data into various major water masses in the study region. Finally, the revised new Figure 10 (formerly Figure 9) illustrates the seasonal evolution of DO in region I corresponding to the changing stratification condition . 3) The figure accompanying the manuscript are very poor and confusing. Again the large quantity of data they collected should deserve a better analysis, based (for instance) on an objective analysis procedure, who would allow the author to define better the location of the water masses. From a formal point of view the combined use of isolines and "colored" dots is adding confusion.

R: The new Figure 9 and revised Figure 10 present objective analyses of the large quantify of data following the suggestion of the reviewer. In Figures 3 and 5-8, the presentation of both color dots and isolines is kept to ease the recognition of spatial gradients by a reader.

4) As it is now it is very much difficult to relate the main conclusion of the paper

with the data described and analysed earlier. This is due, as stated above, to the poor treatment of the data and the generic analysis procedure.

R: After a major revision following the comments and suggestions, we hope that the connection between our analysis and conclusion is more explicit. We thank the reviewer again for the time spent on our manuscript.

Please also note the supplement to this comment:
http://www.nat-hazards-earth-syst-sci-discuss.net/nhess-2016-59/nhess-2016-59-AC1-supplement.pdf

---

## Author Comment (AC2) · 15 Aug 2016

We thank both reviews for very constrictive comments and suggestions. These are very helpful in guiding our revisions of the manuscript. We make the efforts to thoroughly revise in order to improve the presentation of our data analysis and the interpretation of results. A point-to-point response to comments is provided below. Please also note that we have changed the title of this manuscript to "On influencing factors of hypoxia in waters adjacent to the Changjiang estuary" and added an additional author (Youyu Lu).

Reviewer 2: Based on a new and extensive dataset (cruises) the authors intent to relate the dynamics of hypoxic area (formation, spatial distribution, timing) in the Changjiang Estuary to local hydrodynamics features. The strong hypoxic event documented highlights the scientific relevance of the issue. The most important conclusion is that the interannual variability of hypoxic events in this region is primarily driven by physical processes rather than riverine run off or nutrient input (see comparison with Gulf of Mexico in Sect 4.4) and the details provided for this control. The authors details those physical process for two distinct regions, i.e 1. the southern estuary, where hypoxia is avoided in cases of Kuroshio Subsurface water northward extension and 2. the Changjiang Bank (or northern estuary) where hypoxia occurrence is related to persistent (haline) stratification triggered by the westward spreading of Changjiang diluted waters. Those conclusions are well supported by the summarizing Fig. 9. and form a understanding basis for a more extensive reading of local datasets. The analysis is well documented and instructed, in the sense that observations are described and discussed in the lights of know dynamics of the region rather than passed through automatic statistical procedures. In my view this is positive given the complexity of local circulation features. The abstract gives a clear summary of the results and conclusions. I advise the publication of this manuscript, after considering of the few minor revisions listed below.

R: The reviewer provides concise summary of our study and we greatly appreciate the very encouraging comments. Following the reviews of this and the other reviewer, we have significantly revised the manuscript through improving presentation and analysis. We hope the reviewer find the revised manuscript satisfactory.

Major comments. Sect 4.4 I think the discussion should include comments on the different timescales considered. Biogeochemical and physical drivers does not play on the interannual variability of hypoxia occurrences at the same time scales. Another issue is that the gulf of Mexico and the East China Sea differ by their openess, which might changes the relative importance of physical and biogeochemical drivers.

R: Thank you for the suggestion. In the revised section 4.4, we have added discussions on interannual variation of hypoxia and the distinction between the Gulf of Mexico and the East China Sea in terms of the potential influence of their openness.

Minor Comments. L62. "the influence of pycnocline on the spatial variation of hypoxia" -> the relationship between the pycnocline ?location? and spatial variation of hypoxia ?

This sentence is changed to "Zhu et al. (2015) emphasized the occurrence of hypoxia related to the presence of pycnocline".

L70 extra space in "t hat"

Revised.

L73 Can you give reference for the blocking of oxygen exchanges by ocean eddies? if not, remove.

We have removed the mentioning of eddies.

L103 senor -> sensor

Revised.

L111 references needed for the "investigation standards"

A reference has been added.

L112 phycnocline -> pycnocline

Revised.

L148. Either "reached 30m at station K3", or " reached 20-30 m from stations K2 to K6"

This sentence has been rephrased as 'the thickness of the hypoxia water column reached 20-30 m from stations K2 to K6,...'.

L166.and Fig 8 can you give more references or the method used to locate the frontal zone.

More references related to the tidal fronts have been added in Section 2.

L172 oxygenate aeretion -> oxygenation or ventilation

Revised.

L176. Reference for characteristic oxygen concentrations of the KSW.

The related references have been added. We re-check the references and revised the characteristic oxygen concentrations of the KSW to be about 5.0-6.0 mg L-1.

L184 Tsushima strait is not on the map Fig1.

Thanks. It is now on the revised Fig 1.

L191. "was consistent"

Revised.

L201 rephrase "the persistence of stratification was insufficient for oxygen consumption"

This sentence has been changed to 'In May, the cumulative oxygen consumption was insufficient to cause low values of DO, despite of the development of stratification,"

L204 "DO was rapidly consumed" -> "was rapidly depleted".

Revised.

L216 rephrase: "The absence of DO-rich water input from upstream led to .. "

Revised.

L217 "rest" -> "other"

Revised.

L243 remove the uppercase from Log and put N2 inside ()

Revised.

L259 "was far more less" -> "was lower by far"

Revised.

L273 needs -> requires

Revised.

L286 remove "of"

Revised.

L296 involve -> evolve

Revised.

L305 last sentence has to be rephrased to something like "Our conclusions could support the prediction of hypoxia based on the evolution of bottom and surface salinity"

The last sentence has been changed to 'We hope that combining analyses of observational data and modelling results shall eventually lead to better prediction of the spatial and temporal variations of hypoxia.' References : Chen et al is 1988 on L.29 and 1998 in the references; Zhao et al 1985 not reffered to in the manuscript. Revised. Thank the reviewer again for the very critical review.

Please also note the supplement to this comment:
http://www.nat-hazards-earth-syst-sci-discuss.net/nhess-2016-59/nhess-2016-59-AC2-supplement.pdf

**Supplement:**

Response to Review

We thank both reviews for very constrictive comments and suggestions. These are very helpful in guiding our revisions of the manuscript. We make the efforts to thoroughly revise in order to improve the presentation of our data analysis and the interpretation of results. A point-to-point response to comments is provided below. Reviewers' comments are in blue font and our response is in black font. Please also note that we have changed the title of this manuscript to "On influencing factors of hypoxia in waters adjacent to the Changjiang estuary" and added an additional author (Youyu Lu).

Reviewer 1:

The manuscript is based on a series of hydrological surveys in the region adjacent and offshore the Changjiang estuary, an area that suffers from frequent hypoxia/anoxia episodes. The background information, provided by the Authors in the introductory part of the manuscript, inform the reader that hypoxia development is due to the classical triggering factors: enhanced vertical stratification and organic matter accumulation and remineralsation in the lower water column, but they state that the timing of the hypoxia onset, as well as the location of the main hypoxia center is determined by the interplay between the water masses of the region. The main effort of the paper is about an effort to define the spatial and temporal hydrological characteristics underlying development of the hypoxia events as well as their location. I have to say that despite the rich and extensive dataset assembled, the effort is not successful because of a rather confusing analysis and description of the data collected. Therefore, I do not recommend the publication of the manuscript in its present form. Below I list a series of remarks that hopefully the authors might consider in addressing the very serious major revision that the manuscript needs in order to be considered for publication in future.

We thank this reviewer for the very critical review that points out the weakness in our analysis. The detailed comments are fully considered in our revision. (See our response below)

1) The manuscript reads very much as a technical report rather than as a scientific paper. There is a long description of the paper figures that unfortunately does not help very much the reader to understand the following considerations.

Following the reviewer's comment, we have made significant effort to simply description of observational data (figures) and enhance the statistical analysis and interpretation.

2) The hydrology of the region (Water mass distribution and pathways of the main currents) is described by means of a qualitative cartoon only. However, in the following the distribution of the observed hydrological properties (T S and DO) is related to specific water masses. Unfortunately the reader not knowledgeable with the oceanographic characteristics of the region, cannot fully understand and assess the dependence of the hypoxia onset and location on the basis of the changing hydrology. It is therefore strongly recommended that the authors define more strictly the hydrological properties of the water masses involved in the hypoxia dynamics of the region (the large quantity of data they collected should enable them to provide (for instance) T-S diagram whose analysis can help to define in a quantitative way the interplay among water masses. Also the use of T-DO and/or S-DO diagrams could greatly help the analysis and the considerations about hypoxia timing and location.

35     Following this very insightful and constructive suggestion from the reviewer, we have made substantial effort to improve the analysis of relationship between water mass and DO distributions. First, Figure 1 has been revised to denote the major branches of summer circulation that contribute to the shaping of hydrologic conditions in the region. We divide the focused study area into regions I and II, to simplify the description in the main text and ease the appreciation by a reader who is unfamiliar with the regional oceanography. We further add a new Figure 9 to explore the variations of DO with respect to T,

40     S and $N^2$. These new diagrams show clear grouping of observational data into various major water masses in the study region. Finally, the revised new Figure 10 (formerly Figure 9) illustrates the seasonal evolution of DO in region I corresponding to the changing stratification condition.

3) The figure accompanying the manuscript are very poor and confusing. Again the large quantity of data they collected should deserve a better analysis, based (for instance) on an objective analysis procedure, who would allow the author to

45     define better the location of the water masses. From a formal point of view the combined use of isolines and "colored" dots is adding confusion.

The new Figure 9 and revised Figure 10 present objective analyses of the large quantify of data following the suggestion of the reviewer. In Figures 3 and 5-8, the presentation of both color dots and isolines is kept to ease the recognition of spatial gradients by a reader.

50     4) As it is now it is very much difficult to relate the main conclusion of the paper with the data described and analysed earlier. This is due, as stated above, to the poor treatment of the data and the generic analysis procedure.

After a major revision following the comments and suggestions, we hope that the connection between our analysis and conclusion is more explicit. We thank the reviewer again for the time spent on our manuscript.

Based on a new and extensive dataset (cruises) the authors intent to relate the dynamics of hypoxic area (formation, spatial distribution, timing) in the Changjiang Estuary to local hydrodynamics features. The strong hypoxic event documented highlights the scientific relevance of the issue. The most important conclusion is that the interannual variability of hypoxic
60    events in this region is primarily driven by physical processes rather than riverine run off or nutrient input (see comparison with Gulf of Mexico in Sect 4.4) and the details provided for this control. The authors details those physical process for two distinct regions, i.e 1. the southern estuary, where hypoxia is avoided in cases of Kuroshio Subsurface water northward extension and 2. the Changiiang Bank (or northern estuary) where hypoxia occurrence is related to persistent (haline) stratification triggered by the westward spreading of Changjiang diluted waters. Those conclusions are well supported by the
65    summarizing Fig. 9. and form a understanding basis for a more extensive reading of local datasets. The analysis is well documented and instructed, in the sense that observations are described and discussed in the lights of know dynamics of the region rather than passed through automatic statistical procedures. In my view this is positive given the complexity of local circulation features. The abstract gives a clear summary of the results and conclusions. I advise the publication of this manuscript, after considering of the few minor revisions listed below.

70    The reviewer provides concise summary of our study and we greatly appreciate the very encouraging comments. Following the reviews of this and the other reviewer, we have significantly revised the manuscript through improving presentation and analysis. We hope the reviewer find the revised manuscript satisfactory.

Major comments.

Sect 4.4 I think the discussion should include comments on the different timescales considered. Biogeochemical and physical
75    drivers does not play on the interannual variability of hypoxia occurrences at the same time scales. Another issue is that the gulf of Mexico and the East China Sea differ by their openess, which might changes the relative importance of physical and biogeochemical drivers.

Thank you for the suggestion. In the revised section 4.4, we have added discussions on interannual variation of hypoxia and the distinction between the Gulf of Mexico and the East China Sea in terms of the potential influence of their openness.

80    Minor Comments.

 L62. "the influence of pycnocline on the spatial variation of hypoxia" -> the relationship between the pycnocline ?location? and spatial variation of hypoxia ?

This sentence is changed to "Zhu et al. (2015) emphasized the occurrence of hypoxia related to the presence of pycnocline".

L70 extra space in "t hat"

85    Revised.

L73 Can you give reference for the blocking of oxygen exchanges by ocean eddies? if not, remove.

We have removed the mentioning of eddies.

L103 senor -> sensor

Revised.

L111 references needed for the "investigation standards"

A reference has been added.

L112 phycnocline -> pycnocline

Revised.

L148. Either "reached 30m at station K3", or " reached 20-30 m from stations K2 to K6"

This sentence has been rephrased as 'the thickness of the hypoxia water column reached 20-30 m from stations K2 to K6,…'.

L166.and Fig 8 can you give more references or the method used to locate the frontal zone.

More references related to the tidal fronts have been added in Section 2.

L172 oxygenate aeretion -> oxygenation or ventilation

Revised.

L176. Reference for characteristic oxygen concentrations of the KSW.

The related references have been added. We re-check the references and revised the characteristic oxygen concentrations of the KSW to be about 5.0-6.0 mg L$^{-1}$.

L184 Tsushima strait is not on the map Fig1.

Thanks. It is now on the revised Fig 1.

L191. "was consistent"

Revised.

L201 rephrase "the persistence of stratification was insufficient for oxygen consumption"

This sentence has been changed to 'In May, the cumulative oxygen consumption was insufficient to cause low values of DO, despite of the development of stratification,"

L204 "DO was rapidly consumed" -> "was rapidly depleted".

Revised.

L216 rephrase: "The absence of DO-rich water input from upstream led to .. "

Revised.

L217 "rest" -> "other"

Revised.

L243 remove the uppercase from Log and put N2 inside ()

Revised.

L259 "was far more less" -> "was lower by far"

Revised.

L273 needs -> requires

Revised.

L286 remove "of"

Revised.

L296 involve -> evolve

Revised.

L305 last sentence has to be rephrased to something like "Our conclusions could support the prediction of hypoxia based on the evolution of bottom and surface salinity"

The last sentence has been changed to 'We hope that combining analyses of observational data and modelling results shall eventually lead to better prediction of the spatial and temporal variations of hypoxia.'

References : Chen et al is 1988 on L.29 and 1998 in the references; Zhao et al 1985 not reffered to in the manuscript.

Revised. Thank the reviewer again for the very critical review.

**On influencing factors of hypoxia in waters adjacent to the Changjiang estuary**

Xiaofan Luo[1*], Hao Wei[2*], Renfu Fan[2], Zhe Liu[1], Liang Zhao[1], Youyu Lu[3]

[1] College of Marine Science and Engineering, Tianjin University of Science & Technology, Tianjin, 300457, China

[2] School of Marine Science and Technology, Tianjin University, Nankai District, Tianjin, 300072, China

[3] Ocean Sciences Division, Department of Fisheries and Oceans, Bedford Institute of Oceanography, Dartmouth, Nova Scotia, B2Y 4A2, Canada

*Correspondence to*: Hao Wei (weihao@ouc.edu.cn)

*[*]These authors contributed equally to this work and should be considered co-first authors.*

**Abstract.** Based on observational data from ten cruises carried out in 2012 and 2013, the distribution of dissolved oxygen (DO) and hypoxia (DO <2.0 mg L$^{-1}$) evolution in waters adjacent to the Changjiang estuary are studied. The linkage between summer hypoxia and hydrodynamic conditions is explored. The results suggest that hypoxia frequently occurred from June to October to the south of the Changjiang estuary near the 30-50 m isobaths and was prone to happen under strong stratification without the presence of the Kuroshio Subsurface Water (KSW). Over the Changjiang Bank, hypoxia mainly occurred in July, August and September. Low-oxygen areas initially appear under strong stratification induced by the spreading of the Changjiang Diluted Water (CDW), and develop into hypoxia centers due to the lack of supplement of the relatively DO-rich Yellow Sea Water and the KSW. The evolution of hypoxia in a year is influenced by conditions of the shelf circulation especially the paths of the KSW and the CDW. Thus, further study on the salinity evolution in the bottom layer of the water to the south of the Changjiang estuary and in the surface layer over the Changjiang Bank, that indicates the extensions of the KSW and the CDW, is needed for improving the hypoxia prediction.

**1 Introduction**

Most species of marine living depend on dissolved oxygen (DO) in the water and are threatened by low concentration of DO (i.e., <3.0 mg L$^{-1}$). In coastal waters, increasing occurrence of extremely low DO concentration, i.e. hypoxia (DO <2.0 mg L$^{-1}$), is becoming a global environmental issue (Diaz and Rosenberg, 2008; Conley et al., 2009). One of the main goals of regional environment management is to monitor and control the area and volume of hypoxia, and this requires understanding and prediction of low-oxygen evolution (Feng et al., 2012). It is generally agreed that stratification and organic matter degradation are main reasons for the formation of hypoxia. Stratification is influenced by various physical processes. For hypoxia in estuarine regions, there have been continuous debates on the roles played by the riverine nutrient loads and

165    freshwater discharge (Bianchi et al., 2010).

       Since the middle of last century, seasonal survey has revealed the occurrence of hypoxia in the lower water column adjacent to the Changjiang estuary an area with a high primary production on the shelf of East China Sea (ECS) (Fig. 1) (Gu, 1980; Chen et al., 1988; Tian et al., 1993; Zhao et al., 2001). However, hypoxia was not identified as an important factor affecting the ecosystem in this region until the study carried out by the Chinese GLOBEC (Global Ocean Ecosystem

170    Dynamics) project in the summer of 1999 (Li et al., 2002). Subsequently, summer hypoxia in this region has been continuously reported based on sparse ship-based (Shi et al., 2006a; Wei et al., 2007; Wang, 2009). Currently, the Changjiang estuary has been regarded as one of the largest coastal hypoxia areas in the world (Chen et al., 2007).

       A completed picture of the spatial and temporal variations of DO concentration adjacent to the Changjiang estuary has started to emerge based on analysis of observation data collected so far. From May to October in 2006, intense observations

175    were made in this region by various research teams (Zhang et al., 2007; Zhou et al., 2010; Li et al., 2011; Wei et al., 2011; Wang et al., 2012). The synthesis of analyzing observations revealed that in 2006, low-oxygen center did not appear in May but existed in June in waters along the coast to the south of the Changjiang estuary. In July, a hypoxia center firstly appeared in the western part of the Changjiang Bank, and then in August, two hypoxia centers appeared to the north and south of the Changjiang estuary, with the one in the north being more severe. In September and October, the hypoxia center existed to the

[revised manuscript text omitted]

Stations to the south of 33° N are chosen for analyzing hypoxia in water adjacent to the Changjiang estuary (Fig. 1).

250 Sampling grids for each cruise and sections to be discussed in following sections are shown in Fig. 2. According to Su et al. (1996), salinity can be used to identify water mass in the ECS, i.e., $S$ <30.0 representing the diluted water, $S$ >34.0 for the TWC water that includes the KSW with $S$ >34.5. In general, the 17.0 ℃ isothermal can be as the boundary between the YS cold water and its surrounding water (Mao et al., 1964). Thus, we take the area with bottom temperature in the range of 13.0-17.0 ℃ to represent the summer tidal front of the YS (Zhao, 1985). According to China's 1992 National Marine Investigation

255 Standard by the State Bureau of Technical Supervision (the State Bureau of Technical Supervision, 1992), the pycnocline is defined to exist if the vertical gradient of density is larger than 0.1 kg m$^{-4}$ for the water depth less than 200 m. This equals to the maximum $N^2$ of a vertical profile larger than $1.0 \times 10^{-3.0}$ s$^{-2}$ ($N^2$ is used to denote the maximum $N^2$ of a vertical profile in the following).

**3. Bottom DO concentration observed in 2012 and 2013**

**3.1. June, August and October, 2012**

There was no hypoxia in June of 2012 (Fig. 3a). The bottom DO concentration was larger than 4.0 mg L$^{-1}$, with higher values in the eastern part and lower values in the western part of survey area. A band of relatively low DO concentration of 4.0-5.0 mg L$^{-1}$ was found along the 30 m isobath to the north of 29° N.

In August (Fig. 3b), the DO concentration was 2.0-3.0 mg L$^{-1}$ lower than that in June. There were only a few stations located in Region II. Over the western side of Region I, DO <3.0 mg L$^{-1}$ was observed. A hypoxia center with DO = 1.0 mg L$^{-1}$ appeared in the northwestern survey field (station I1, at 33° N, 122.5° E).

In October, the DO concentration of bottom water was elevated (Fig. 3c). In Region II, two centers of relatively low-oxygen were present. These two centers were located near the northern end of the submarine river canyon with DO = 4.8 mg L$^{-1}$ (station PN1), and the steep slope with DO <4.0 mg L$^{-1}$, about 2.0 mg L$^{-1}$ lower than that in the surrounding waters.

Overall, in summer of 2012 hypoxia off the Changjiang estuary was not severe. There was no hypoxia in June and October though low-oxygen centers existed. The hypoxia center was located in Region I in August.

**3.2. May - September, 2013**

In May, the vertical and horizontal distributions of DO were fairly homogenous (Fig. 3d, vertical distribution was not shown). DO values were 8.0-9.0 mg L$^{-1}$ in the near shore region and 7.0-8.0 mg L$^{-1}$ in the offshore region. The minimum DO value, located at station kb, was 6.7 mg L$^{-1}$.

In June, DO values at most stations in Region I were less than 5.0 mg L$^{-1}$ (Fig. 3e). In Region II, DO was lower than 4.0 mg L$^{-1}$ and a hypoxia center appeared near the slope between the 30-50 m isobaths. Along the 122.5° E section (black lines in Fig. 2a, e), the DO concentration in the lower water column in Region I (31° N - 33.5° N) was substantially larger than that in Region II (28.5° N - 31° N) where the hypoxia water had a thickness of about 10 m (Fig. 4b). In addition, bottom DO in June of 2013 was overall about 2.0-3.0 mg L$^{-1}$ lower than that in June of 2012 (Figs. 3a, e and Figs. 4a, b), and hypoxia occurred earlier in 2013. Previously, the occurrence of hypoxia in June was only reported in 2003 in the submarine river canyon near 30.45° N (Xu, 2005).

In July, bottom DO was larger than 7.0 mg L$^{-1}$ in the Changjiang estuary and Hangzhou Bay, except that DO <4.0 mg L$^{-1}$ was found at stations X5 and X6 near the northern corner of the Changjiang estuary (Fig. 3f). In Region I, at all stations with depths larger than 30 m, bottom DO was less than 3.0 mg L$^{-1}$; severe hypoxia occurred within 32-32.4° N, 122.4-123.5° E over the western part of Region I. The huge area with DO <3.0 mg L$^{-1}$ extended eastward. Over the middle shelf of the ECS, DO values were 3.0-5.0 mg L$^{-1}$. Overall, the DO concentrations in the southern and eastern parts of the survey area were higher than that in the middle part.

In August, the survey area was smaller than that in July (Fig. 3g). In Region I, the area of hypoxia was still large, and extended northward and eastward compared with July; DO <2.0 mg L$^{-1}$ was observed at five stations and hypoxia area was

separated into several patches extending northward to 33° N and eastward to 125° E. Along section K (at 32° N, green lines in Fig. 2b, g), DO distribution showed that the thickness of the hypoxia water column reached 20-30 m from stations K2 to K6, extending about 100 kilometer in length (Fig. 4d), with a much lower DO value than that in August 2012 (Fig. 4c). In Region II, a low-oxygen center with DO <3.0 mg L$^{-1}$ was located in the northern end of the submarine river canyon.

In September, bottom DO over the western Region I increased quickly, while DO <3.0 mg L$^{-1}$ was still found near the northeastern edge of this region and hypoxia occurred near 33° N (Fig. 3h). Another area with DO <3.0 mg L$^{-1}$ was in the river canyon in Region II.

In 2013, hypoxia first appeared in Region II in June and was sustained in Region I in July, August and September. Multiple low-oxygen centers appeared from June to September. In Region II, the hypoxia centers were stably located in the coastal water near the 30-50 m isobaths. In Region I, the positions of hypoxia centers changed around the bank. Overall, compared with historical reports, in the summer of 2013 hypoxia appeared earlier, occupied a larger area, with a larger thickness, and was maintained longer over the Changjiang Bank.

**4. Discussion on influencing factors of DO concentration**

**4.1 Hydrodynamic conditions based on observations in 2012 and 2013**

Figures 5-8 present the spatial distribution and temporal evolution of $N^2$, bottom salinity, surface salinity and bottom temperature observed from the 2012 and 2013 cruises. The hydrographic conditions in the two years show a general consistency with the climatology based on historical observations (Editorial Board for Marine Atlas-Hydrology, 1992), but also show notable year-to-year differences. Spatial and temporal variations of hydrography in this region are influenced by bathymetry, surface forcing associated with monsoon, runoff of the Changjiang River, and circulation patterns including Kuroshio, TWC, YSC and coastal currents (Su, 2001).

The northward TWC consists of Taiwan Strait water in the upper layer characterized with high temperature and moderately high salinity ($S$ >34.0), and the KSW in the lower layer with low temperature and high salinity ($S$ >34.5) (Weng and Wang, 1985). Region II is the main region passing through by the TWC. The presence of stratification in this region is largely due to the occupation of the TWC water, and is also influenced by the coastal fresh water ($S$ <31.0) (Fig. 5 and Fig. 6). The KSW can be clearly identified by the characteristic oxygen concentration of about 5.0-6.0 mg L$^{-1}$ (Pan et al., 1993; Liu et al., 2012). Hence the KSW provides the relatively low and high oxygen water before and after the formation of low-oxygen center (DO <3.0 mg L$^{-1}$), respectively. In 2012, the KSW intruded northward over 31° N in June and then reached at the north corner of the Changjiang estuary in August, covering the whole Region II and southern part of Region I (Fig. 6a-b). Till October, the KSW retreated (Fig. 6c). While in 2013, the intrusion of KSW was much weaker than that in 2012 (Fig. 6). The most northward intrusion occurred in July of 2013 (Fig. 6f).

In Region I, a strong halocline is created by the fresher CDW ($S$ <30.0) in the upper layer and the saltier lower layer

water. The CDW extension is mainly influenced by the advancement and recession of the TWC, especially its inner branch that flows parallel to the 50 m isobath into the river canyon and reaches the northern corner of the Changjiang estuary (Weng and Wang, 1985; Wei et al., 2015a). When the TWC intrudes further north over the continental shelf, the CDW tends to veer northeastward substantially (Wei et al., 2015a). This is the situation that occurred in 2012 (Fig. 6a-c and Fig. 7a-c). In 2013, with a weaker northward intrusion of the TWC, and combined with strong southerly wind (Wei et al., 2015a), the CDW expanded eastward significantly in July and resulted in strong stratification in the whole Region I (Fig. 5f and 7f).

The tidal front in the YS appears in May corresponding to surface heating and gets intensified gradually in the following three months. In Region I, this front is usually located near the northeastern edge and extends southeastward (Fig. 8). The YSW with relatively high DO concentration is transported by the YSC into Region I along the east side of the front. Here we use bottom $T$ <15.0 ℃ to identify this water mass. In addition, the current along the western coast of the YS within 20 m isobath (i.e., SBCC in Fig. 1) flows northward as a response to the wind in summer (Liu and Hu, 2009). Both tidal front and northward coastal current are main obstacles for supplement of the YSW to the low-oxygen water in most parts of Region I.

In summary, the hydrodynamic condition, especially the TWC and the CDW, plays a leading role in the evolution, duration and intensity of stratification in waters adjacent to the Changjiang estuary. According to laboratory experiments, Liu et al. (2012) proposed that the main reason for hypoxia formation was not oxygen consumption but the lack of replenishment after consumption. Hence hypoxia may occur in any area if stratification is sustained over a considerable duration without DO replenishment.

**4.2. CDW spreading and stratification in Region I**

In Region I, the low-oxygen centers are located where stratification is strong, and the pattern of hypoxia is influenced by the extension of the CDW.

In June 2012, both strong stratification and relative low-oxygen occurred in the western part of this region (Figs. 3a, 5a); and till August, strong stratification was maintained and DO decreased to less than 3.0 mg L$^{-1}$ (Figs. 3b, 5b). The CDW veered northeastward substantially (Fig. 7b), and below it a salinity front was produced as the TWC encountered the coastal fresh water (Fig. 6b). To the east of salinity front, there existed a tidal front (Fig. 8b). The DO-rich YSW was transported to the eastern area along this tidal front, leaving the absence of DO replenishment to the north of salinity front in western part. This resulted in the development of the previous low-oxygen area into hypoxia (Fig. 3b). In October, the CDW switched back to flowing southward along the coast. As stratification disappeared ($N^2$ <10$^{-3.5}$ s$^{-2}$), bottom DO increased to 7.0 mg L$^{-1}$ (Figs. 3c, 5c).

In summer of 2013, the low-oxygen center and hypoxia over the Changjiang Bank all showed correspondence with strong stratification ($N^2$ >10$^{-2.0}$ s$^{-2}$) (Fig. 3 and Fig. 5). In July, the CDW expanded eastward, covering a zone spanning from the Changjiang estuary to Jeju Island. This facilitated the formation of a large low-oxygen area with DO <3.0 mg L$^{-1}$ over the

whole bank under strong stratification (Figs. 3f, 5f, 7f). In the meanwhile, DO supplement from the KSW and the YSW
inhibited the formation of hypoxia in the southern and eastern bank, respectively (Figs. 6f, 8f). Hence, hypoxia events only
happened in the western bank without DO supplement (Fig. 3f). In August, the KSW did not approach the Changjiang
estuary, but retreated southward (Fig. 6g). Meanwhile, the northward migration of the tidal front (Fig. 8g) limited the DO
replenishment by the YSW. As a consequence, a sever hypoxia over the eastern bank evolved from the low-oxygen with
sustained stratification (Fig. 3g, 5g). In September, the CDW shrunk, and stratification was broken almost everywhere (Figs.
5h, 7h) with DO being increased (Fig. 3h). Near the northeastern edge of Region I, however, stratification ($N^2$ >$10^{-3.0}$ s$^{-2}$) was
still persisted because of the vertical temperature differences, and low DO (<3.0 mg L$^{-1}$) was maintained (stations I4, I5 and
K6) with hypoxia occurring at the northeastern location (station I5).

   In the bottom water of Region I, the YSW ($T$ <15.0 °C), CDW ($S$ <30.0), TWC water ($S$ >34.0) and KSW ($S$ >34.5) can
be clearly identified, while the water of other salinity ranges is regarded as the Shelf Mixed Water (SMW) (Fig. 9a).
Generally, the YSW in the north and KSW in the south provide relatively DO-rich water into this region. Thus, hypoxia was
unlikely to happen in areas occupied by the YSW and KSW, but usually appeared in the SMW (Fig. 9a). This indicates that
there is a limitation to track the hypoxia development based on bottom T and S. The relationship of DO with surface salinity
and stratification (Fig. 9c) shows overall lower DO values and the occurrence of hypoxia under the CDW in summer.
Hypoxia happened only when stratification was strong, while DO values were always larger than 4.0 mg L$^{-1}$ when
stratification was very weak ($N^2$ <$10^{-3.0}$ s$^{-2}$). Hypoxia and high DO concentration could both happen under strong
stratification, suggesting that stratification is a necessary but insufficient condition for the formation of hypoxia over the
bank. Despite of this, the bottom DO concentration in Region I with depths greater than 30 m was overall negatively
correlated with the strength of stratification in summer and autumn, with a regression relationship of DO = -1.7 × log$_{10}$($N^2$) +
0.4 (r = -0.7, significant at the 0.05 confidence level) (Fig. 10a). In 2013, low-oxygen area was initially formed under the
presence of strong stratification associated with the CDW. Then DO decreased continuously with the persistence of
stratification even though the stratification weakened slightly (Fig. 10c). In areas without the supplement of DO-rich water,
low-oxygen evolved into hypoxia (Fig. 10c). In 2012, the pattern of DO evolution was essentially the same as that in 2013
(Fig. 10b). When stratification was totally broken, DO recovered to a higher value (Fig. 10b).

   In summer, the CDW spreads over the bank but the orientation of its extension may vary. For instance, the CDW spread
northward in June 2012, and eastward in July 2013. The largest values of $N^2$ were reached when the CDW lay over salty
waters. Overall, hypoxia usually formed over the northwestern Changjiang Bank when the CDW extended northwestward,
and in the eastern part when the CDW spread eastward substantially.

**4.3. KSW intrusion and stratification in Region II**

   In Region II, the appearance of low-oxygen centers is consistent with the distribution of strong stratification (Fig. 5), and the
evolution of DO concentration can be mainly influenced by the lateral transport of the KSW.

A low-oxygen center often first occurs in the confluent area of coastal current (i.e., MZCC in Fig. 1) and the TWC, which is usually located between 30 m and 50 m isobaths where stratification is stronger and more persistent than that in the middle shelf of the ECS. In June and August of 2012, because of the intensive KSW intrusion toward the northern corner of the Changjiang estuary (Figs. 6a, b), hypoxia did not happen despite of the presence of strong stratification ($N^2 > 10^{-2.5}$ s$^{-2}$) (Figs. 5a, b). In October of 2012, owing to the TWC recession and enhanced monsoon, stratification was overall weak and DO increased (Figs. 3c, 5c, 6c).

From May to September of 2013, the coastal area was strongly stratified (Figs. 5d-h). In May, the cumulative oxygen consumption was insufficient to cause low values of DO, despite of the development of stratification. In June, the southern branch of the CDW expanded more widely relative to June 2012 (Figs. 7a, e), and the TWC reached the latitude of 30.5° N (Fig. 6e). This led to large values of $N^2$ ($N^2 > 10^{-2.0}$ s$^{-2}$) in the coastal area of Region II (Fig. 5e). With the presence of sustained stratification (May to June), DO was rapidly depleted and a large low-oxygen area formed. In the meanwhile, the KSW was located to the south of 28° N (Figs. 6e, 8e) and could not provide the relatively DO-rich water. As a consequence, low-oxygen developed into hypoxia (Fig. 3e). In July, with the KSW intruding northward and occupying the majority of bottom water in region II (Fig. 6f), hypoxia faded away and did not re-appear afterward. Thus, the timely replenishment of the KSW could prevent the evolution of low-oxygen into hypoxia.

The bottom water of Region II is mainly composed of TWC water and KSW, and with some additions of Coastal Water (CW: $S$ <31.0) and SMW (Fig. 9b). The distribution of DO against bottom T-S (Fig. 9b) indicates that hypoxia did not occur when $S$ >34.5, i.e., in the KSW; and when hypoxia occurred, $S$ must be less than 34.5. The hypoxic potential was likely created by the TWC water due to the sustained stratification, and hypoxia tended to evolve in the absence of the KSW. There are evidences of this happened in the past. For example, previous studies reported the occurrence of hypoxia in the coastal area of Region II defined by us in August of 1959, 1976-85, 1981, 1999, 2002 and 2006; June of 2003; and September and October of 2006 (Table 1). Among these years, in 2006 hypoxia was maintained for three months while the KSW intruded at a southern location (Zou et al., 2008; Zhou et al., 2010; Wang et al., 2012); and in August 1959 hypoxia occurred in the northern end of the canyon where bottom salinity was 33.0 (Liu et al., 2012). For the other hypoxia cases, there was insufficient hydrological data to identify the locations of the KSW intrusion.

**4.4. An inter-comparison with hypoxia in the Gulf of Mexico**

Worldwide, the frequent occurrence of hypoxia has been related to eutrophication (Conley et al., 2009). For the Gulf of Mexico, a statistical model for hypoxia prediction has been established based on nutrient loads from Mississippi River (Turner et al., 2006). The interannual variation of the extent of hypoxia waters has a strong correlation with the volume of river discharge in the northern Gulf of Mexico in spring and summer (Rabouille et al., 2008). However, for waters adjacent to the Changjiang estuary, hypoxia can be influenced by many complicated factors. Here hypoxia is not directly related to runoff from river. In summer of 2006 the runoff was extremely low (Fig. 11) but severe hypoxia occurred (Zou et al., 2008;

Zhou et al., 2010; Wang et al., 2012). In summer of 2012 the runoff was among the highest in the recent decade but only a small area of hypoxia existed in August (Figs. 3a-c and Fig. 11). Compared with 2012, in 2013 the runoff was lower by far but hypoxia occupied extremely large areas and sustained much longer (Figs. 3d-h and Fig. 11). Clearly, the area or volume of hypoxia adjacent to the Changjiang estuary cannot be predicted based on river runoff.

Over the Texas-Louisiana shelf in the northern Gulf of Mexico, the formation and destruction of hypoxia is primarily a local vertical process, and the biological processes are responsible for the variation of hypoxia from east to west (Hetland and DiMarco, 2008; Bianchi et al., 2010). Hypoxia is predominantly caused by water column respiration in the eastern region with steep shelf break, and by benthic respiration over the wide shelf to the west. Similarly, the hypoxia area adjacent to the Changjiang estuary can be divided into an area on the wide bank (Region I) and an area with steep slope (in Region II) according to bathymetry. Over the Changjiang Bank, low-oxygen is related to the presence of the CDW, while the CDW's position is influenced by the TWC (Wei et al., 2015a). Strong tidal mixing and monsoon may both induce the detachment of freshwater from the CDW (Xuan et al., 2012). This results in stronger changes of the river plume on the Changjiang Bank, compared with that on the west part of the Texas-Louisiana shelf where tides and wind are relatively weak. In Region II of the ECS, the occurrence of low-oxygen centers near the steep slope is related to bathymetric features, and the bottom DO in this region is significantly influenced by the shelf circulation including the TWC (stratification) and shoreward intrusion of the Kuroshio (DO replenishment).

Circulation conditions in the norther Gulf of Mexico and the ECS are distinctly different. The absence of strong offshore currents in the Mississippi margin results in a long residence time of the bottom water, in favor of biological respiration and hence frequent occurrence of hypoxia over a large spatial scale (Rabouille et al., 2008). Circulation in the ECS is strongly influenced by the presence of the strong western boundary current (Kuroshio). This results in a shorter residence time of the bottom water and hence less regular occurrence of hypoxia (Rabouille et al., 2008). Thus, in water adjacent to the Changjiang estuary, physical processes dominate over biological processes in driving the evolution of hypoxia. The prediction of low-oxygen requires salinity distribution at the bottom in Region II and at the surface in Region I, that defines the extensions of the KSW and the CDW, respectively.

**5. Conclusions**

As one of the major causes of ecological disasters in the coastal seas, especially in large river estuaries, hypoxia is becoming a global environmental issue and has attracted wide attention in recent decades. Base on new observational data from 10 cruises carried out in 2012 and 2013, the distribution of dissolved oxygen and evolution of hypoxia in waters adjacent to the Changjiang estuary are studied. The linkage of summer hypoxia with hydrodynamic conditions, including variations of water mass and stratification associated with circulation, is explored. The study area can be divided into Region I (the Changjiang Bank) and Region II (the region to the south of the estuary). The mechanisms dominating the evolution of stratification and DO distribution are different in these two regions.

a) In Region I, i.e., over the Changjiang Bank, there is a high possibility that hypoxia occurs from July to September. The distribution of bottom DO corresponds to the intensity of stratification, the two being in negative correlation according to DO = $-1.7 \times \log_{10}(N^2) + 0.4$. Hypoxia exists only when stratification is strong, while DO is larger than 4.0 mg L$^{-1}$ when stratification is weak ($N^2 < 10^{-3.0}$ s$^{-2}$). Over the bank, the appearance of low-oxygen areas is mostly related to the spreading of the Changjiang Diluted Water. The combination of the coastal current (SBCC in Fig.1, northward in summer) and tidal front restrains DO replenishment from surrounding waters. Therefore, under the condition of sustained stratification, low-oxygen tends to evolve into hypoxia. Generally, the coastal current transports the Yellow Sea Water to the Changjiang Bank after the transition of monsoon in autumn, when stratification has already disappeared. Thus, the inflow of the YSW may play a minor role in the relief of summer hypoxia over the bank.

[revised manuscript text omitted]

585  [a] YECS: Yellow and East China Seas

590

595 Table 3. Averaged values of DO (in mg L$^{-1}$) and the logarithm of the maximum squared buoyancy frequency (in s$^{-2}$) in Region I from different monthly cruises. $N$ denotes the number of samples.

| cruise | DO | $N^2$ | N |
|--------|------|---------|----|
| 201206 | 6.5±1.6 | -2.3±0.9 | 12 |
| 201208 | 3.9±1.1 | -2.0±0.5 | 10 |
| 201210 | 7.1±0.4 | -3.9±0.3 | 11 |
| 201306 | 4.3±0.7 | -2.5±0.5 | 11 |
| 201307 | 2.6±0.6 | -1.8±0.3 | 16 |
| 201308 | 2.3±0.7 | -2.1±0.4 | 13 |
| 201309 | 4.8±1.9 | -2.8±0.8 | 14 |

[Figure]

600

Figure 1. Topography and schematic map of summer circulation in the Yellow Sea and East China Sea. Black arrows denote circulation including the Yellow Sea Current (YSC), Subei Coastal Current (SBCC), Minzhe Coastal Current (MZCC), Taiwan Warm Current (TWC), Kuroshio Subsurface Branch Current (KSBC) and Kuroshio. CE, HB and RC are short for the Changjiang estuary, Hangzhou Bay and submarine river canyon, respectively. Various symbols represent sampling

605  stations of different cruises labelled in the right pane that observed the occurrence of hypoxia. Polygons outlined by green lines are I. Changjiang Bank and II. the region to the south of the Changjiang estuary. The Contours in gray denote isobaths of 30, 50, 100, and 200 m. Dotted blue line marks the boundary between the Yellow sea and the East China Sea.

610

[Figure]

**Figure 2.** Maps of sampling stations off the Changjiang estuary (a-c: June, August and October in 2012; d-h: May to September in 2013). The layout is designed to ease comparison between 2012 and 2013 for June and August. Black, green and blue bold lines denote section at 122.5°E and section K in Fig. 4, and section PN in Fig. 5, respectively.

[Figure]

620

**Figure 3.** Dissolved oxygen concentration (mg L$^{-1}$) of bottom water in the same layout as Fig. 2.

[Figure]

625 **Figure 4.** Dissolved oxygen concentration (mg L$^{-1}$) along section at 122.5°E (a: June 2012; b: June 2013) and section K (c: August 2012; d: August 2013).

[Figure]

630

**Figure 5.** Same as Fig. 3 except for maximum squared buoyancy frequency at logarithmic scale ($N^2$: s$^{-2}$).

[Figure]

635

**Figure 6.** Same as Fig. 3 except for bottom salinity (in psu). The contour of $S = 30.0$ in blue represents the extension of the Changjiang Diluted Water; $S = 34.0$ in magenta represents the Taiwan Warm Current; and $S = 34.5$ in red indicates the intrusion of Kuroshio Subsurface Water.

[Figure]

**Figure 7.** Same as Fig. 6 except for surface salinity (in psu).

640

[Figure]

**Figure 8.** Same as Fig. 3 except for bottom temperature (in °C). Shading area denotes frontal zone.

[Figure]

Figure 9. Variations of DO (in mg L$^{-1}$) corresponding to bottom salinity (in psu) and bottom temperature (°C) for (a) Region I and (b) Region II, and (c) corresponding to surface salinity (in psu) and the logarithm of maximum squared buoyancy frequency ($N^2$: in s$^{-2}$) for Region I. Dots stand for samplings collected in summer (June to August); diamonds in spring (May) and autumn (September and October). TWCW, KSW, SMW, YSW, CDW and CW are abbreviations of Taiwan Warm Current Water, Kuroshio Subsurface Water, Shelf Mixed Water, Yellow Sea Water, Changjiang Diluted Water and Coastal Water.

[Figure]

Figure 10. The distribution of DO (in mg L$^{-1}$) versus $\log_{10}$ ($N^2$) in Region I with water depth greater than 30 m for

660    (a) all samplings in 2012 and 2013 and separately in (b) 2012 and (c) 2013. Color symbols denote samplings collected during different monthly cruises. The black solid line in (a) denotes the linear regression relationship. In (b) and (c), black arrows illustrate the seasonal evolution of DO in 2012 and 2013, respectively; colored dots denote the averaged values of DO and $N^2$ of different months (also listed in Table 3).

665

[Figure]

670  **Figure 11.** Monthly discharge of the Changjiang River in 2000–2013, averaged from daily monitoring data collected at the
Datong Hydrologic Station (http://yu-zhu.vicp.net/).

**On influencing factors of hHypoxia  in waters adjacent to the Changjiang estuary**

Xiaofan Luo[1*], Hao Wei[2*], Renfu Fan[2], Zhe Liu[1], Liang Zhao[1], Youyu Lu[3]

[1] College of Marine Science and Engineering, Tianjin University of Science & Technology, Tianjin, 300457, China

[2] School of Marine Science and Technology, Tianjin University, Nankai District, Tianjin, 300072, China

[3] Ocean Sciences Division, Department of Fisheries and Oceans, Bedford Institute of Oceanography, Dartmouth, Nova Scotia, B2Y 4A2, Canada

*Correspondence to*: Hao Wei (weihao@ouc.edu.cn)

*[*]These authors contributed equally to this work and should be considered co-first authors.*

**Abstract.** Based on observational data from ten cruises carried out in 2012 and 2013, the distribution of dissolved oxygen (DO) and hypoxia (DO <2.0 mg L$^{-1}$) evolution in waters adjacent to the Changjiang estuary are studied. The linkage between summer hypoxia and hydrodynamic conditions is explored. The results suggest that hypoxia frequently occurred from June to October to the south of the Changjiang estuary near the 30-50 m isobaths and was prone to happen under strong stratification without the presence of the Kuroshio Subsurface Water (KSW). Over the Changjiang Bank, hypoxia mainly occurred in July, August and September. Low-oxygen areas initially  appear under strong stratification induced by the spreading of the Changjiang Diluted Water (CDW), and develop into hypoxia centers due to the lack of supplement of the relatively DO-rich Yellow Sea Water and the KSW. The evolution of hypoxia in a year is influenced by conditions of the shelf circulation especially the paths of the KSW and the CDW. Thus, further study on the salinity evolution in the bottom layer of the water to the south of the Changjiang estuary and in the surface layer over the Changjiang Bank, that indicates the extensions of the KSW and the CDW, is needed for improving the hypoxia prediction.

**1 Introduction**

Most species of marine living depend on dDissolved oxygen (DO) in the water and are threatened by

 low concentration of DO (i.e., $<3.0$ mg L$^{-1}$) . In coastal waters, increasing occurrence of extremely low DO concentration, i.e. hypoxia (DO $<2.0$ mg L$^{-1}$), is becoming a global environmental issue (Diaz and Rosenberg, 2008; Conley et al., 2009). One of the main goals of regional environment management is to  monitor and control the area and volume of hypoxia, and this requires understanding and prediction of low-oxygen evolution (Feng et al., 2012). It is generally agreed that stratification and organic matter degradation are main reasons for the formation of hypoxia. Stratification is influenced by  various physical processes.  For hypoxia in estuarine regions, there have been continuous debates on the roles played by the riverine nutrient loads and freshwater discharge (Bianchi et al., 2010).

Since the middle of last century, seasonal survey has revealed the occurrence of hypoxia in the lower water  column adjacent to the Changjiang estuary , an area with  high primary production on the shelf of East China Sea (ECS) (Fig. 1) (Gu, 1980; Chen et al., 1988; Tian et al., 1993; Zhao et al., 2001). However, hypoxia was not identified as an important factor affecting the ecosystem  in this region until the study carried out by the Chinese GLOBEC (Global Ocean Ecosystem Dynamics) project in the summer of 1999 (Li et al., 2002). Subsequently, summer hypoxia in this region has been continuously reported based on sparse ship-based observations  (Shi et al., 2006a; Wei et al., 2007; Wang, 2009). Currently, the Changjiang estuary has been regarded as one of the largest coastal hypoxia areas in the world (Chen et al., 2007).

A completed picture of the spatial and temporal variations of DO concentration adjacent to the Changjiang estuary has started to emerge based on analysis of observation data collected so far.  From May to October in 2006, intense observations were made in this region by various  research teams (Zhang et al., 2007; Zhou et al., 2010; Li et al., 2011; Wei et al., 2011; Wang et al., 2012). The synthesis of analyzing observations revealed that In 2006, low-oxygen center did not appear in May but existed in June in waters along the coast to the south of the Changjiang estuary. In July, a hypoxia center firstly appeared in the western part of the Changjiang Bank, and then in August, two hypoxia centers appeared to the north and south of the  Changjiang estuary , with the one in the north being more severe. In September and October, the hypoxia center existed to the south of the Changjiang estuary. Data from the monthly survey from September of 1958 to September of 1959 was recently re-analyzed (Liu et al., 2012). It was found that the low-oxygen center appeared to the south of the estuary in May and June, appeared near the  estuary in July and August, and re-appeared to the south of the estuary in September. This evolution of the low-oxygen center showed correspondence with the extension of the Taiwan Warm Current (TWC) which consists of Taiwan Strait water in the upper layer and the Kuroshio Subsurface Water (KSW) in the lower layer. The lowest DO value was 0.34 mg L$^{-1}$ in August of 1959, a highly severe hypoxia but did not draw much attention for about 40 years.

[revised manuscript text omitted]

Stations to the south of 33° N are chosen for analyzing hypoxia in water adjacent to  the Changjiang estuary (Fig. 1). Sampling grids for each cruise and sections to be discussed in following sections are shown in Fig. 2. According to Su et al. (1996), salinity can be used to identify water mass in the ECS, i.e., $S$ <30.0 representing the diluted water, $S$ >34.0 for the TWC water that includes the KSW with $S$ >34.5 . In general, the 17.0 ℃ isothermal can be as the boundary between the YS cold water and its surrounding water (Mao et al., 1964). Thus,  we take the area with bottom temperature in the range of 13.0-17.0 ℃  to represent the summer tidal front of the YS (Zhao, 1985). According to China's 1992 National Marine Investigation Standard by the State Bureau of Technical Supervision (the State Bureau of Technical Supervision, 1992), the pycnocline is defined to exist if the vertical gradient of density is larger than 0.1 kg m$^{-4}$ for the water depth less than 200 m. This equals to the maximum $N^2$ of a vertical profile larger than 1.0×10$^{-3.0}$ s$^{-2}$ ( $N^2$ is used to denote the maximum $N^2$ of a vertical profile in the following).

**3. Bottom DO concentration observed in 2012 and 2013**

**3.1.  June, August and October, 2012**

There was no hypoxia in June of 2012 (Fig. 3a). The bottom DO concentration was larger than 4.0 mg L$^{-1}$, with higher values in the eastern part and lower values in the western part of survey area. A band of relatively low DO concentration of 4.0-5.0 mg L$^{-1}$ was found along the 30 m isobath to the north of 29° N.

In August (Fig. 3b), the DO concentration was 2.0-3.0 mg L$^{-1}$ lower than that in June. There were only a few stations located in Region II . . Over the western side of Region I, DO <3.0 mg L$^{-1}$ was observed. A hypoxia center with DO = 1.0 mg L$^{-1}$ appeared in the northwestern survey field (station I1, at 33° N, 122.5° E).

In October, the DO concentration of bottom water was elevated (Fig. 3c). In Region II, two centers of relatively low-oxygen were present . These two centers were located near the northern end of the submarine river canyon with DO = 4.8 mg L$^{-1}$ (station PN1), and the steep slope with DO <4.0 mg L$^{-1}$, about 2.0 mg L$^{-1}$ lower than that in the surrounding waters.

Overall, in summer of 2012 hypoxia off the Changjiang estuary was not severe. There was no hypoxia in June and October though  low-oxygen centers existed. The hypoxia center was located in Region I in August.

**3.2.  May  - September, 2013**

In May, the vertical and horizontal distributions of DO were fairly homogenous (Fig. 3d, vertical distribution was not shown).

DO values were 8.0-9.0 mg L$^{-1}$ in the near shore region and 7.0-8.0 mg L$^{-1}$ in the offshore region. The minimum DO value, located at station kb, was 6.7 mg L$^{-1}$.

In June, DO values at most stations in Region I were less than 5.0 mg L$^{-1}$ (Fig. 3e). In Region II, DO was lower than 4.0 mg L$^{-1}$  and a hypoxia center appeared near the slope  between the 30-50 m isobaths  Along the 122.5° E section (black lines in Fig. 2a, e), the DO concentration in the lower water column  in Region I (31° N - 33.5° N) was substantially larger than that  in Region II (28.5° N - 31° N) where the hypoxia water had a thickness of about 10 m (Fig. 4b).  In addition, bottom DO in June of 2013 was overall about 2.0-3.0 mg L$^{-1}$ lower than that in June of 2012 (Figs. 3a, e and Figs. 4a, b), and hypoxia occurred earlier in 2013. Previously, the occurrence of hypoxia in June was only reported in 2003 in the submarine river canyon near 30.45° N (Xu, 2005).

In July, bottom DO was larger than 7.0 mg L$^{-1}$ in the Changjiang estuary and Hangzhou Bay, except that DO <4.0 mg L$^{-1}$ was found at stations X5 and X6 near the northern corner of the Changjiang  estuary (Fig. 3f).  In Region I, at all stations with  depths larger than 30 m, bottom DO was less than 3.0 mg L$^{-1}$; severe hypoxia occurred within 32-32.4° N, 122.4-123.5° E over the western part of  Region I. The huge area with DO <3.0 mg L$^{-1}$ extended  eastward. Over the middle shelf of the ECS, DO values were 3.0-5.0 mg L$^{-1}$ . Overall, the DO concentrations in the southern and eastern parts of the survey area were higher than that in the middle part.

In August, the survey area was smaller than that in July (Fig. 3g).  In Region I, the area of hypoxia was still large, and extended northward and eastward compared with July ; DO <2.0 mg L$^{-1}$ was observed at five stations and hypoxia area was separated into several patches extending northward to 33° N and eastward to 125° E. Along section K (at 32° N, green lines in Fig. 2b, g), DO distribution showed that the thickness of the hypoxia water column reached 20-30 m from stations K2 to K6, extending about 100 kilometer in length (Fig. 4d), with a much lower DO value than that in August 2012 (Fig. 4c). In  Region II, a low-oxygen center with DO <3.0 mg L$^{-1}$ was located in the northern end of the submarine river canyon. ~~DO distribution along section K (at 32° N) showed that the hypoxia water reached 30 m thickness from stations K2 to K6 over the Changjiang Bank, extending about 100 km in length (Fig. 4d). DO <2.0 mg L$^{-1}$ was observed at five stations and hypoxia area was separated into several patches extending northward to 33° N and eastward to 125° E. Over the whole bank, DO was less than 4.0 mg L$^{-1}$, much lower than that in August 2012 (Fig. 4c).~~

In September,  bottom DO over the western  Region I increased quickly, while DO <3.0 mg L$^{-1}$ was still found near  the  northeastern edge of this region  and hypoxia occurred near 33° N (Fig. 3h). Another area with DO <3.0 mg L$^{-1}$ was in the river canyon  in Region II.

In 2013, hypoxia first appeared in  Region II in June and was sustained

860 in Region I in July, August and September. Multiple low-oxygen centers appeared from June to September. In Region II, the hypoxia centers were stably located in the coastal water near the 30-50 m isobaths. In Region I, the positions of hypoxia centers changed around the bank. Overall, compared with historical reports, in the summer of 2013 hypoxia appeared earlier, occupied a larger area, with a larger thickness, and was maintained longer over the Changjiang Bank.

**4. Discussion on influencing factors of DO concentration**

**4.1 Hydrodynamic conditions based on observations in 2012 and 2013**

Figures 5-8 present the spatial distribution and temporal evolution of $N^2$, bottom salinity, surface salinity and bottom temperature observed from the 2012 and 2013 cruises. The hydrographic conditions in the two years show a general consistency with the climatology based on historical observations (Editorial Board for Marine Atlas Hydrology, 1992), but also show notable year-to-year differences. Spatial and temporal variations of hydrography in this region  are influenced by  bathymetry, surface forcing associated with monsoon, runoff of the Changjiang River, and circulation patterns including Kuroshio, TWC, YSC and coastal currents  (Su, 2001).

The northward TWC consists of Taiwan Strait water in the upper layer characterized with high temperature and moderately high salinity ($S$ >34.0), and the KSW in the lower layer with low temperature and high salinity ($S$ >34.5) (Weng and Wang, 1985). Region II is the main region passing through by the TWC. The presence of stratification in this region is largely due to the occupation of the TWC water, and is also influenced by the coastal fresh water ($S$ <31.0) (Fig. 5 and Fig. 6). The KSW can be clearly identified by the characteristic oxygen concentration of about 5.0-6.0 mg L$^{-1}$ (Pan et al., 1993; Liu et al., 2012). Hence the KSW provides the relatively low and high oxygen water before and after the formation of low-oxygen center (DO <3.0 mg L$^{-1}$), respectively. In 2012, the KSW intruded northward over 31° N in June and then reached at the north corner of the Changjiang estuary in August, covering the whole Region II and southern part of Region I (Fig. 6a-b). Till October, the KSW retreated (Fig. 6c). While in 2013, the intrusion of KSW was much weaker than that in 2012 (Fig. 6). The most northward intrusion occurred in July of 2013 (Fig. 6f).

In Region I, a strong halocline is created by the fresher CDW ($S$ <30.0) in the upper layer and the saltier lower layer water. The CDW extension is mainly influenced by the advancement and recession of the TWC, especially its inner branch that flows parallel to the 50 m isobath into the river canyon and reaches the northern corner of the Changjiang estuary (Weng and Wang, 1985; Wei et al., 2015a). When the TWC intrudes further north over the continental shelf, the CDW tends to veer northeastward substantially (Wei et al., 2015a). This is the situation that occurred in 2012 (Fig. 6a-c and Fig. 7a-c). In 2013, with a weaker northward intrusion of the TWC, and combined with strong southerly wind (Wei et al., 2015a), the CDW expanded eastward significantly in July and resulted in strong stratification in the whole Region I (Fig. 5f and 7f).

In each year, stratification starts to develop in May corresponding to surface heating. The tidal front in the YS appears in May corresponding to surface heating and gets intensified gradually in the following three months. In Region I, This this front is usually located near the northeastern edge can and extends southeast-ward across the Changjiang Bank(Fig. 8). The YSW with relatively high DO concentration is transported by the YSC into Region I along the east side of the front. Here we use bottom $T$ <15.0 ℃ to identify this water mass. In summeraddition, the current along the western coast of the YS (within 20 m isobath) (i.e., SBCC in Fig. 1) flows northward as a response to the wind in summer (Liu and Hu, 2009). Both tidal front and northward coastal current are main obstacles for supplement of the YSW to the low-oxygen water over the Changjiang Bankin most parts of Region I.

The CDW extension is mainly influenced by the advancement and recession of the TWC, especially its inner branch (Weng and Wang, 1985; Wei et al., 2015a). When the TWC intrudes further north over the continental shelf, the CDW tends to veer northeastward substantially. A strong halocline over the Changjiang Bank is created by the CDW and the water in lower layer. The combination of halocline and thermal stratification blocks the oxygenate aeration of the lower layer from the surface.

The TWC consists of Taiwan Strait water in the upper layer characterized with high temperature and moderately high salinity, and the KSW in the lower layer with low temperature and high salinity (Weng and Wang, 1985). The pycnocline is formed in the region passing through by the TWC, and is also influenced by the coastal fresh water to the south of the Changjiang estuary. The KSW can be clearly identified by the characteristic oxygen concentration (4.5 mg L$^{-1}$). Hence the KSW provides the relatively low and high oxygen water before and after the formation of low-oxygen center (DO <3.0 mg L$^{-1}$), respectively. The TWC has two branches over the continental shelf. In May, the TWC is reinforced by the southerly wind. The near shore branch of the TWC flows northward along the 50-60 m isobath offshore of Zhejiang into the submarine river canyon, and in July it can reach the northern corner of the Changjiang river mouth. This branch upwells to the sea surface in the coastal area to the south of the Changjiang estuary. The outer branch of the TWC can reach the Changjiang Bank along the middle shelf of the ECS. It flows along the edge of the bank in May and June, across the bank in July and August, along the edge again in September and October. Eventually, the TWC runs into the Tsushima Strait (Su, 2001).

In summary, the hydrodynamic condition, especially the TWC and the CDW, plays a leading role in the evolution, duration and intensity of stratification in waters adjacent to the Changjiang estuary. According to laboratory experiments, Liu et al. (2012) proposed that the main reason for hypoxia formation was not oxygen consumption but the lack of replenishment after consumption. Hence hypoxia may occur in any area if stratification is sustained over a considerable duration without DO replenishment.

**4.2. Bottom DO, CDW spreading and stratification in Region I**

In Region I, the low-oxygen centers are located where stratification is strong, and the pattern of hypoxia is influenced by the extension of the CDW.

In June 2012, both strong stratification and relative low-oxygen occurred in the western part of this region (Figs. 3a, 5a); and till August, strong stratification was maintained and DO decreased to less than 3.0 mg L$^{-1}$ (Figs. 3b, 5b). The CDW veered northeastward substantially (Fig. 7b), and below it a salinity front was produced as the TWC encountered the coastal fresh water (Fig. 6b). To the east of salinity front, there existed a tidal front (Fig. 8b). The DO-rich YSW was transported to the eastern area along this tidal front, leaving the absence of DO replenishment to the north of salinity front in western part. This resulted in the development of the previous low-oxygen area into hypoxia (Fig. 3b). In October, the CDW switched back to flowing southward along the coast. As stratification disappeared ($N^2$ <10$^{-3.5}$ s$^{-2}$), bottom DO increased to 7.0 mg L$^{-1}$ (Figs. 3c, 5c).

In summer of 2013, the low-oxygen center and hypoxia over the Changjiang Bank all showed correspondence with strong stratification ($N^2$ >10$^{-2.0}$ s$^{-2}$) (Fig. 3 and Fig. 5). In July, the CDW expanded eastward, covering a zone spanning from the Changjiang estuary to Jeju Island. This facilitated the formation of a large low-oxygen area with DO <3.0 mg L$^{-1}$ over the whole bank under strong stratification (Figs. 3f, 5f, 7f). In the meanwhile, DO supplement from the KSW and the YSW inhibited the formation of hypoxia in the southern and eastern bank, respectively (Figs. 6f, 8f). Hence, hypoxia events only happened in the western bank without DO supplement (Fig. 3f). In August, the KSW did not approach the Changjiang estuary, but retreated southward (Fig. 6g). Meanwhile, the northward migration of the tidal front (Fig. 8g) limited the DO replenishment by the YSW. As a consequence, a sever hypoxia over the eastern bank evolved from the low-oxygen with sustained stratification (Fig. 3g, 5g). In September, the CDW shrunk, and stratification was broken almost everywhere (Figs. 5h, 7h) with DO being increased (Fig. 3h). Near the northeastern edge of Region I, however, stratification ($N^2$ >10$^{-3.0}$ s$^{-2}$) was still persisted because of the vertical temperature differences, and low DO (<3.0 mg L$^{-1}$) was maintained (stations I4, I5 and K6) with hypoxia occurring at the northeastern location (station I5).

In the bottom water of Region I, the YSW ($T$ <15.0 °C), CDW ($S$ <30.0), TWC water ($S$ >34.0) and KSW ($S$ >34.5) can be clearly identified, while the water of other salinity ranges is regarded as the Shelf Mixed Water (SMW) (Fig. 9a). Generally, the YSW in the north and KSW in the south provide relatively DO-rich water into this region. Thus, hypoxia was unlikely to happen in areas occupied by the YSW and KSW, but usually appeared in the SMW (Fig. 9a). This indicates that there is a limitation to track the hypoxia development based on bottom T and S. The relationship of DO with surface salinity and stratification (Fig. 9c) shows overall lower DO values and the occurrence of hypoxia under the CDW in summer. Hypoxia happened only when stratification was strong, while DO values were always larger than 4.0 mg L$^{-1}$ when stratification was very weak ($N^2$ <10$^{-3.0}$ s$^{-2}$). Hypoxia and high DO concentration could both happen under strong stratification, suggesting that stratification is a necessary but insufficient condition for the formation of hypoxia over the bank. Despite of this, the bottom DO concentration in Region I with depths greater than 30 m was overall negatively correlated with the strength of stratification in summer and autumn, with a regression relationship of DO = -1.7 × log$_{10}$($N^2$) + 0.4 (r = -0.7, significant at the 0.05 confidence level) (Fig. 10a). In 2013, low-oxygen area was initially formed under the presence of strong stratification associated with the CDW. Then DO decreased continuously with the persistence of

stratification even though the stratification weakened slightly (Fig. 10c). In areas without the supplement of DO-rich water, low-oxygen evolved into hypoxia (Fig. 10c). In 2012, the pattern of DO evolution was essentially the same as that in 2013 (Fig. 10b). When stratification was totally broken, DO recovered to a higher value (Fig. 10b).

960   In summer, the CDW spreads over the bank but the orientation of its extension may vary. For instance, the CDW spread northward in June 2012, and eastward in July 2013. The largest values of $N^2$ were reached when the CDW lay over salty waters. Overall, hypoxia usually formed over the northwestern Changjiang Bank when the CDW extended northwestward, and in the eastern part when the CDW spread eastward substantially.

**4.3. KSW intrusion and stratification in Region II**

965   In Region II, the appearance of low-oxygen centers  consistent with the distribution of strong stratification (Fig. 5), and the evolution of DO concentration can be mainly influenced by the lateral transport of the KSW .

A low-oxygen center often first occurs in the confluent area of coastal current (i.e., MZCC in Fig. 1) and the TWC,
which is usually located between 30 m and 50 m isobaths where stratification is stronger and more persistent than that in the
970   middle shelf of the ECS. In June and August of 2012, because of the intensive KSW intrusion toward the northern corner of the Changjiang estuary (Figs. 6a, b), hypoxia did not happen despite of the presence of strong stratification ($N^2 > 10^{-2.5}$ s$^{-2}$) (Figs. 5a, b). In October of 2012, owing to the TWC recession and enhanced monsoon, stratification was overall weak and DO increased (Figs. 3c, 5c, 6c).

From May to September of 2013, the  coastal area was strongly stratified (Figs. 5d-h ). In May, the
975   cumulative oxygen consumption was insufficient to cause low values of DO, despite of the development of stratification. In June, the southern branch of the CDW expanded more widely relative to June 2012 (Figs. 7a, e), and the TWC reached the latitude of 30.5° N (Fig. 6e). This led to  large values of $N^2$  ($N^2 > 10^{-2.0}$ s$^{-2}$) in the coastal  area of Region II (Fig. 5e). With the presence of sustained stratification (May to June), DO
980   was rapidly  depleted and a large low-oxygen area formed. In the meanwhile, the KSW was located to the south of 28° N (Figs. 6e, 8e) and could not provide the relatively DO-rich water . As a consequence,  low-oxygen  developed into hypoxia (Fig. 3e). In July, with the KSW intruding northward and occupying the majority of bottom water in  region II (Fig. 6f), hypoxia faded away and did not re-appear afterward. Thus, the timely replenishment of the KSW could prevent the evolution of low-oxygen  into hypoxia.

985   The bottom water of Region II is mainly composed of TWC water and KSW, and with some additions of Coastal Water
(CW: $S < 31.0$) and SMW (Fig. 9b). The distribution of DO against bottom T-S  (Fig. 9b) indicates that hypoxia did not occur when $S > 34.5$, i.e., in the KSW; and when hypoxia occurred, $S$ must be less than 34.5. The hypoxic potential was likely created by the TWC water due

to the sustained stratification, and hypoxia tended to evolve in the absence of the KSW. . There are evidences of this happened in the past.  For example, previous studies reported the occurrence of hypoxia  in the coastal area of Region II defined by us  in August of 1959, 1976-85, 1981, 1999, 2002 and 2006; June of 2003; and September and October of 2006 (Table 1). Among these years, in 2006 hypoxia was maintained for three months while the KSW intruded at a southern location (Zou et al., 2008; Zhou et al., 2010; Wang et al., 2012); and in August 1959 hypoxia occurred in the northern end of the canyon where bottom salinity was 33.0 (Liu et al., 2012). . For the other hypoxia cases, there was insufficient hydrological data to identify the locations of the KSW intrusion.

[revised manuscript text omitted]

In Region II of the ECS,  the occurrence of low-oxygen centers near the steep slope  is related to  bathymetric features, and the bottom DO in this region  is significantly influenced by the shelf circulation including the TWC (stratification) and shoreward intrusion of the Kuroshio (DO replenishment).

Circulation conditions in the northern Gulf of Mexico and the ECS are distinctly different. The absence of strong offshore currents in the Mississippi margin results in a long residence time of the bottom water, in favor of biological respiration and hence frequent occurrence of hypoxia over a large spatial scale (Rabouille et al., 2008). Circulation in the ECS is strongly influenced by the presence of the strong western boundary current (Kuroshio). This results in a shorter residence time of the bottom water and hence less regular occurrence of hypoxia (Rabouille et al., 2008). Thus, in water adjacent to the Changjiang estuary, physical processes dominate over biological processes in driving the evolution of hypoxia. The prediction of low-oxygen  requires  salinity distribution  at the bottom  in Region II and at the surface  in Region I, that defines the extensions of the KSW and the CDW, respectively.

**5. Conclusions**

As one of the major causes of ecological disasters in the coastal seas, especially in large river estuaries, hypoxia is becoming a global environmental issue and has attracted wide attention in recent decades. Base on new observational data from 10 cruises carried out in 2012 and 2013, the distribution of dissolved oxygen and evolution of hypoxia in waters adjacent to the Changjiang estuary are studied. The linkage of summer hypoxia with hydrodynamic conditions, including variations of water mass and stratification associated with circulation, is explored. The study area can be divided into  Region I (the Changjiang Bank)  and  Region II (the region to the south of the estuary) . The mechanisms dominating the evolution of stratification and DO distribution are different in these two regions.

a) In Region I, i.e., over the Changjiang Bank, there is a high possibility that hypoxia occurs from July to September. The distribution of bottom DO corresponds to the intensity of stratification, the two being in negative correlation according to DO = -1.7 × $\log_{10}(N^2)$ + 0.4. Hypoxia exists only when stratification is strong, while DO is larger than 4.0 mg L$^{-1}$ when stratification is weak ($N^2$ <10$^{-3.0}$ s$^{-2}$). Over the bank, the appearance of low-oxygen areas is mostly related to the spreading of the Changjiang Diluted Water. The combination of the coastal current (SBCC in Fig.1, northward in summer) and tidal front restrains DO replenishment from surrounding waters. Therefore, under the condition of sustained stratification, low-oxygen tends to evolve into hypoxia. Generally, the coastal current transports the Yellow Sea Water to the Changjiang Bank after the transition of monsoon in autumn, when stratification has already disappeared. Thus, the inflow of the YSW may play a minor role in the relief of summer hypoxia over the bank.

b) In  Region II, a confluent area of coastal current and the Taiwan Warm Current near the 30-

[revised manuscript text omitted]

a offshore waters of Zhejiang Province and submarine river canyon

b over the Changjiang Bank with water depth less than 50 m

| cruise | DO | $N^2$ | N |
|---|---|---|---|
| 201206 | 6.5±1.6 | -2.3±0.9 | 12 |
| 201208 | 3.9±1.1 | -2.0±0.5 | 10 |
| 201210 | 7.1±0.4 | -3.9±0.3 | 11 |
| 201306 | 4.3±0.7 | -2.5±0.5 | 11 |
| 201307 | 2.6±0.6 | -1.8±0.3 | 16 |
| 201308 | 2.3±0.7 | -2.1±0.4 | 13 |

| | | | |
|---|---|---|---|
| 201309 | 4.8±1.9 | -2.8±0.8 | 14 |

1235

1240

circulation including the Yellow Sea Current (YSC), Subei Coastal Current (SBCC), Taiwan Warm Current (TWC), and Kuroshio intrusion (KBC: surface Kuroshio Branch Current; O-KBBC: offshore Kuroshio Bottom Branch Current; N-KBBC: Nearshore Kuroshio Bottom Branch Current, i.e. Kuroshio Subsurface Water). Contours in gray denote isobaths of 30, 50 70, 100, and 200 m. Various symbols represent sampling stations of different cruises labelled in the right pane that observed the occurrence of hypoxia. Red line marks the boundary between southern and northern sub regions. Sea.

1245

Figure 1. Topography and schematic map of summer circulation in the Yellow Sea and East China Sea. Black arrows denote circulation including the Yellow Sea Current (YSC), Subei Coastal Current (SBCC), Minzhe Coastal Current (MZCC), Taiwan Warm Current (TWC), Kuroshio Subsurface Branch Current (KSBC) and Kuroshio. CE, HB and RC are short for the Changjiang estuary, Hangzhou Bay and submarine river canyon, respectively. Various symbols represent sampling

1250

stations of different cruises labelled in the right pane that observed the occurrence of hypoxia. Polygons outlined by green lines are I. Changjiang Bank and II. the region to the south of the Changjiang estuary. The Contours in gray denote isobaths of 30, 50, 100, and 200 m. Dotted blue line marks the boundary between the Yellow sea and the East China Sea.

1255

[Figure]

**Figure 2.** Maps of sampling stations off the Changjiang estuary (a-c: June, August and October in 2012; d-h: May to September in 2013). The layout is designed to ease comparison between 2012 and 2013 for June and August. Black, green and blue bold lines denote section at 122.5°E and section K in Fig. 4, and section PN in Fig. 5, respectively.

[Figure]

**Figure 3.** Dissolved oxygen concentration (mg L⁻¹) of bottom water in the same layout as Fig. 2.

1265

[Figure]

**Figure 4.** Dissolved oxygen concentration (mg L$^{-1}$) along section at 122.5°E (a: June 2012; b: June 2013) and section K (c: August 2012; d: August 2013).

1270

[Figure]

**Figure 5.** Same as Fig. 3 except for maximum squared buoyancy frequency at logarithmic scale ($N^2$: s$^{-2}$).

[Figure]

1275

**Figure 6.** Sar⋯⋯⋯⋯⋯ity (in psu). ⋯⋯⋯tour of $S$ ⋯⋯⋯⋯⋯⋯on of the Changjiang Diluted Water; $S = 34.0$ in magenta represents the Taiwan Warm Current; and $S = 34.5$ in red indicates the intrusion of Kuroshio Subsurface Water.

1280

[Figure]

**Figure 7.** Same as Fig. 6 except for surface salinity (in psu).

[Figure]

1285

**Figure 8.** Same as Fig. 3 except for bottom temperature (in °C).  Shading area denotes frontal zone.

[Figure]

**Figure 9.**

[Figure]

Figure 9. Variations of DO (in mg L⁻¹) corresponding to bottom salinity (in psu) and bottom temperature (°C) for (a) Region I and (b) Region II, and (c) corresponding to surface salinity (in psu) and the logarithm of maximum squared buoyancy frequency ($N^2$: in s⁻²) for Region I. Dots stand for samplings collected in summer (June to August); diamonds in spring (May) and autumn (September and October). TWCW, KSW, SMW, YSW, CDW and CW are abbreviations of Taiwan Warm

[Figure]

Figure 10. The distribution of DO (in mg L$^{-1}$) versus $\log_{10}$ ($N^2$) in Region I with water depth greater than 30 m for (a) all samplings in 2012 and 2013 and separately in (b) 2012 and (c) 2013. Color symbols denote samplings collected during different monthly cruises. The black solid line in (a) denotes the linear regression relationship. In (b) and (c), black arrows illustrate the seasonal evolution of DO in 2012 and 2013, respectively; colored dots denote the averaged values of DO and $N^2$ of different months (also listed in Table 3).

[Figure]

**Figure 1011.** Monthly discharge of the Changjiang River in 2000–2013, averaged from daily monitoring data collected at the Datong Hydrologic Station (http://yu-zhu.vicp.net/).

1315